# A Bayesian model of context-sensitive value attribution

**Francesco Rigoli[1]\*, Karl J Friston[1], Cristina Martinelli[2], Mirjana Selaković[3], Sukhwinder S Shergill[2], Raymond J Dolan[1,4]**

[1]The Wellcome Trust Centre for Neuroimaging, University College London, London, United Kingdom; [2]Department of Psychosis Studies, Institute of Psychiatry, Psychology and Neuroscience, King's College London, London, United Kingdom; [3]Department of Psychiatry, Sismanoglio General Hospital, Athens, Greece; [4]Max Planck UCL Centre for Computational Psychiatry and Ageing Research, London, United Kingdom

**Abstract** Substantial evidence indicates that incentive value depends on an anticipation of rewards within a given context. However, the computations underlying this context sensitivity remain unknown. To address this question, we introduce a normative (Bayesian) account of how rewards map to incentive values. This assumes that the brain inverts a model of how rewards are generated. Key features of our account include (i) an influence of prior beliefs about the context in which rewards are delivered (weighted by their reliability in a Bayes-optimal fashion), (ii) the notion that incentive values correspond to precision-weighted prediction errors, (iii) and contextual information unfolding at different hierarchical levels. This formulation implies that incentive value is intrinsically context-dependent. We provide empirical support for this model by showing that incentive value is influenced by context variability and by hierarchically nested contexts. The perspective we introduce generates new empirical predictions that might help explaining psychopathologies, such as addiction.

**\*For correspondence:** f.rigoli@ucl.ac.uk

**Competing interests:** The authors declare that no competing interests exist.

## Introduction

Choice preferences vary as a function of context, and recent studies have shed light on the processes underlying these contextual influences (*Huber et al., 1982*; *Johnson and Busemeyer, 2005*; *Ludvig et al., 2013*; *Louie et al., 2013*, *2014*, *2015*; *Roe et al., 2011*; *Rigoli et al., 2016a*, *2016b*; *Simonson and Tversky, 1992*; *Soltani et al., 2012*; *Stewart, 2009*; *Stewart et al., 2003*; *Summerfield and Tsetsos, 2012*, *2015*; *Tsetsos et al., 2010*, *2012*, *2016*; *Tversky, 1972*; *Usher and McClelland, 2004*; *Vlaev et al., 2012*). For example, in (*Rigoli et al., 2016a*, *2016b*) participants performed a task where blocks of trials were associated with either a low or a high-value context (with overlapping distributions). Choice behaviour was consistent with the hypothesis that the incentive values of identical rewards were *larger* in the *low* compared to the high-value context. This and similar evidence suggests that, at least in some cases, contextual effects on choice behaviour are explained by an incentive value that reflects the *relative* value of rewards anticipated within a given context (*Louie et al., 2013*; *Ludvig et al., 2013*; *Rigoli et al., 2016a*; *Stewart, 2009*; *Stewart et al., 2003*).

However, the computational mechanisms underlying the context sensitive nature of incentive value remain unclear. A promising explanatory framework builds on the notion that the brain's computations correspond to Bayesian inference and learning. Several empirical and theoretical arguments support a Bayesian inference as a general account of brain function (*Chater et al., 2006*; *Clark, 2013*; *Dayan et al., 1995*; *Ernst, 2006*; *Friston, 2010*). The application of similar principles

to value learning and planning has inspired the notions of planning as inference and active inference (*Botvinick and Toussaint, 2012*; *Friston et al., 2013*, *2015*; *Pezzulo and Rigoli, 2011*; *Pezzulo et al., 2015*; *Solway and Botvinick, 2012*). Here, we consider the possibility that the context-sensitive value is a product of Bayesian inference. This implies that incentive value will depend on expectation and uncertainty about rewards, conditioned upon contextual factors. If this is the case, we would expect to see choice behaviour change with any contextual variable that is an *ancestor* of rewards in the subject's generative model of these rewards.

We refer to our account as a Bayesian model of context-sensitive value (BCV). Below, we introduce the model and compare it with previous accounts of contextual influence on incentive value and choice (Bushong et al., unpublished; *Kőszegi and Rabin, 2006*; *Kőszegi and Szeidl, 2013*; *Louie et al., 2014*, *2015*). We then report data from two behavioural experiments where we analysed two key predictions of BCV.

## Results

### Bayesian account of context-sensitive value

The general framework of BCV is Bayesian, building on a proposal that the brain performs some form of Bayesian inference (*Chater et al., 2006*; *Clark, 2013*; *Dayan et al., 1995*; *Ernst, 2006*; *Friston, 2010*). This approach considers the brain to possess a generative model of the sensorium, comprising a set of random variables (i.e., hidden states or causes of sensory outcomes) and their causal links (i.e., probabilistic contingencies). The variables can be separated into hidden and observable variables; the former representing the latent causes of observations, and the latter representing sensory evidence or cues. Sensory evidence is conveyed by observable variables, and this evidence is combined with prior beliefs to produce a posterior belief about the (hidden) causes of observations. The application of this logic to perception is straightforward and has proved effective in explaining several empirical phenomena in perception (*Chater et al., 2006*; *Clark, 2013*; *Dayan et al., 1995*; *Ernst, 2006*; *Friston, 2010*). For instance, there is evidence for integrating different perceptual modalities (e.g., visual and tactile) in a manner consistent with Bayesian principles (*Ernst, 2006*).

We propose a Bayesian scheme for BCV that accommodates the influence of context on incentive value. BCV focuses on scenarios (i) where incentive value depends on contextual information (either represented by cues or by previous rewards) provided before options or rewards are presented, and (ii) where reward is defined by a single attribute (e.g., reward amount). To describe the basic principles of BCV, we adopt the formalism of Bayesian graphs (*Bishop, 2006*) where a generative model is described by nodes or circles, representing random variables (shaded and white circles refer to observed and non-observed variables respectively), and arrows, representing causal relationships among variables. A simple generative model hypothesized by BCV is shown in *Figure 1A*, where C represents prior beliefs about the average reward expected in a given context. Formally, this corresponds to a (Gaussian) prior belief (with mean $\mu_c$ and variance $\sigma_c^2$ over the mean of a (Gaussian) distribution of reward options R (with variance $\sigma_R^2$). When R is observed, a posterior expectation about the context is obtained by application of Bayes rule (*Bishop, 2006*):

$$\mu_{C|R} = \mu_C + \frac{\sigma_C^2}{\sigma_C^2 + \sigma_R^2}(R - \mu_C) \qquad (1)$$

The crucial proposal we advance here is that the incentive value V(R) attributed to a certain reward option is embedded within this belief update process and corresponds to a precision-weighted prediction error (*Friston, 2005*); namely, to the difference between R and the expected reward $\mu_C$, multiplied by a gain term, which depends on the variances of both reward and context (i.e., relative confidence or precision):

$$V(R) = \frac{\sigma_C^2}{\sigma_C^2 + \sigma_R^2}(R - \mu_C) \qquad (2)$$

The notion that incentive value corresponds to precision-weighted prediction error links with ideas in other cognitive domains proposing that prior expectations are explained away and perception corresponds to (precision-weighted) residuals, or prediction errors (*Blakemore et al., 1999*;

*Brown et al., 2013*; *Friston, 2005*; *Rao and Ballard, 1999*). For example, it has been proposed that our sense of agency emerges from explaining away action-dependent somatosensory predictions, and hence what is perceived as externally generated sensation corresponds to (precision-weighted) residuals of the sensory input (*Brown et al., 2013*).

Here, we propose that a similar mechanism is involved in attribution of incentive value. This implies two fundamental forms of contextual normalization. First, a subtractive normalization is exerted when $\mu_C$ is different from zero. For example, if we assign positive and negative numbers to rewards and punishments respectively, their corresponding incentive values may change in sign, depending on whether punishment (i.e., $\mu_C < 0$) or reward (i.e., $\mu_C > 0$) is expected on average within the context. This implies that small rewards can appear as losses in contexts where large rewards are expected. Second, a divisive normalization depends on the gain $\frac{\sigma_C^2}{\sigma_C^2 + \sigma_R^2}$. This implies that the positive and negative value of profits (i.e., $R > \mu_C$) and losses (i.e., $R < \mu_C$) will be augmented or attenuated, depending upon the relative precision of prior beliefs about (prior confidence in) the context ($\sigma_C^2$) and of sensory evidence about the reward option ($\sigma_R^2$).

Equation two applies every time novel information about reward is provided, which is when a prediction error occurs. This happens (i) when a (primary or secondary) reward is delivered (or is not delivered when expected), which can be post choice as well as in other conditions (e.g., in classical conditioning paradigms, when a reward is delivered independent of action), and (ii) when one (or more) option is presented. The latter follows because an agent has an expectation about an option, which leads to a prediction error when the actual option is presented.

A key aspect of our proposal addresses how contextual variables are implemented within a generative model. One possibility, illustrated in *Figure 1B,* is a generative model that includes an observation O reporting information about context. This model assumes that a value C is drawn from a Gaussian distribution with mean $\mu_C = 0$ and variance $\sigma_C^2$. A zero mean captures context-independent information, as it implies that overall rewards (i.e., $R > 0$) and punishments (i.e., $R < \mu_C$) will be attributed positive and negative incentive values respectively. The variance $\sigma_C^2$ reflects prior uncertainty about the hidden or latent context. A context observation O is sampled from a Gaussian distribution with mean $f(C)$ and variance $\sigma_O^2$ (reflecting the reliability of the context-related cue). For simplicity, we assume that $f(C) = C$, though in general this can be any function (similar simplifications are assumed below). A reward R is observed after being sampled from a Gaussian distribution with mean C and variance $\sigma_R^2$ (reflecting uncertainty about the reward distribution).

We propose that agents form posterior beliefs about the context P(C|O,R) using Bayesian belief updating – first accumulating contextual information by estimating P(C|O), and then reward information to give P(C|O,R). This sequential inference (c.f., evidence accumulation) is motivated by the fact that information about context is usually provided at an earlier time point than reward options. The mean of the posterior distribution P(C|O) is:

$$\mu_{C|O} = \frac{\sigma_C^2}{\sigma_C^2 + \sigma_O^2} O \tag{3}$$

And the posterior variance:

$$\sigma_{C|O}^2 = \sigma_C^2 - \frac{\sigma_C^2}{\sigma_C^2 + \sigma_O^2} \sigma_C^2 \tag{4}$$

The mean of the posterior distribution P(C|O,R) corresponds to:

$$\mu_{C|O,R} = \mu_{C|O} + \frac{\sigma_{C|O}^2}{\sigma_{C|O}^2 + \sigma_R^2} \left( R - \mu_{C|O} \right) \tag{5}$$

Implying the following incentive value for reward:

$$V(R) = \frac{\sigma_{C|O}^2}{\sigma_{C|O}^2 + \sigma_R^2} \left( R - \mu_{C|O} \right) \tag{6}$$

This shows that, other things being equal, information about the context (reflected in the value of O) induces a subtractive value normalization.

A possible extension of this generative model is illustrated in *Figure 1C* where contexts are organized hierarchically (*Pezzulo et al., 2015*). Imagine evaluating the same dish in different restaurants (a low-level context) and in different neighbourhoods (a high-level context). This example highlights the fact that some (high-level) contexts are more generic, while other (low-level) contexts are more specific. Crucially, if a context exerted no impact on incentive value, one would expect that the dish would be equally attractive, irrespective of where it was experienced. If one context exerted an influence, we would expect, for example, that the incentive value of the dish depends on the restaurant and not on the neighbourhood. Finally, if both contextual levels are in play, one would predict that different incentive values would be attributed to the same dish as a function of both the restaurant

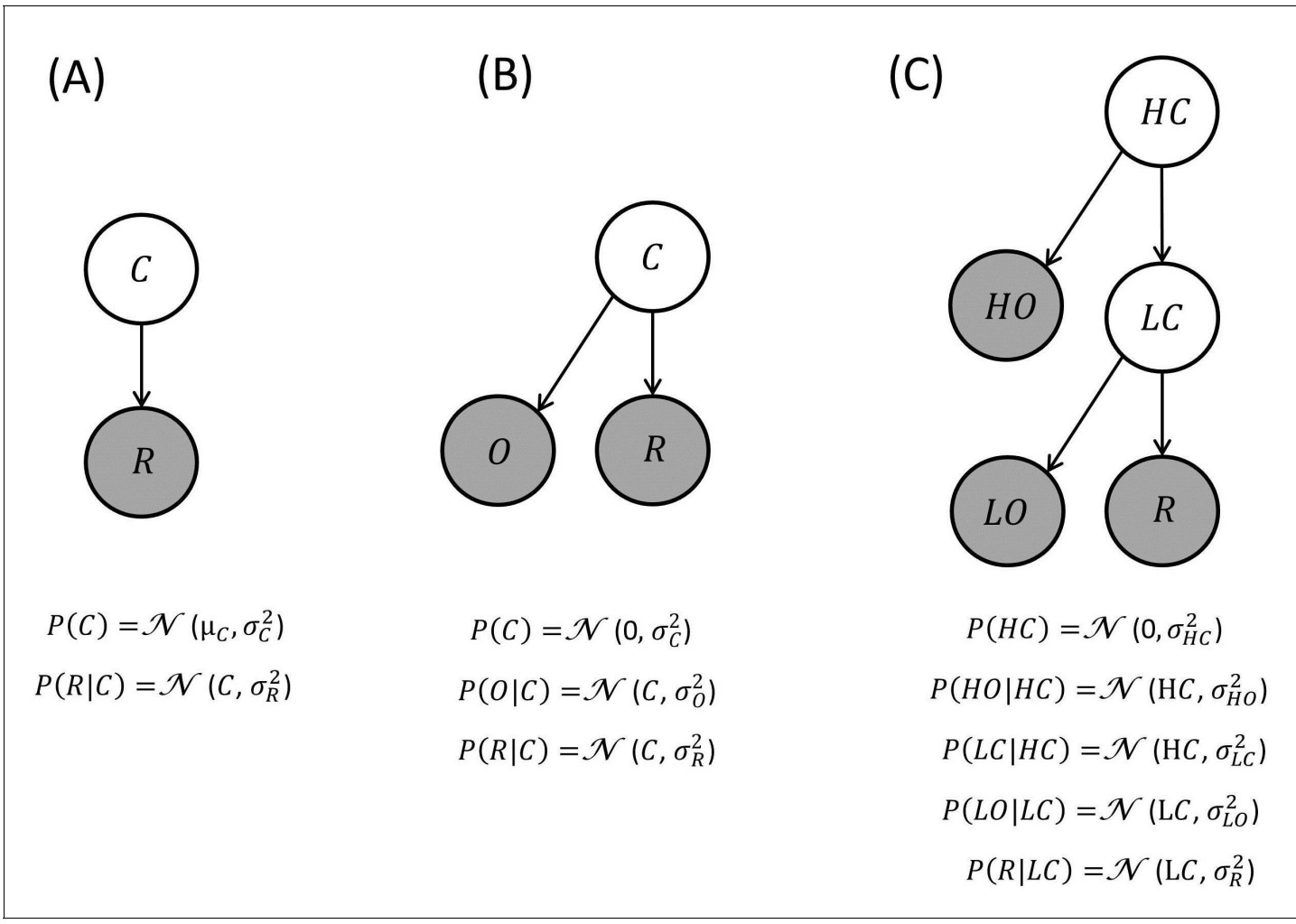

**Figure 1.** Generative models of reward: these generative models are depicted as directed acyclic graphs or Bayesian networks. Circles represent random variables (shaded and white circles refer to observed and non-observed variables respectively). An arrow denotes a conditional dependence – in which one random variable supplies a sufficient statistic of the probability distribution of its children. In BCV, contextual variables generate the sufficient statistics (expectation and variance) of a Gaussian observable variable corresponding to reward. In these examples, the contextual variables generate first-order sufficient statistics of their descendants (e.g., the mean of Gaussian distributions) as in parametric or empirical Bayesian models. Alternatively, the contextual variables could determine the variance of Gaussian random variables; in which case this would be a hierarchical Gaussian filter. Inverting this model, given observations, furnishes posterior beliefs over the context variables. This inference determines incentive value which is conceived as precision-weighted prediction error. (A) Generative model where a contextual variable C reflects a prior belief over the reward mean. (B) Generative model where a contextual variable C generates a prior expectancy of zero over the reward mean, and a noisy observation O of the context is provided. (C) Generative model where context is organized hierarchically and comprises a high level (HC; e.g., a neighbourhood) and a low level (LC; e.g., a restaurant), both associated with noisy observations (HO and LO respectively).

and neighbourhood. Here, we examined the possibility that incentive value depends on generative models where contexts are nested hierarchically. A higher-level contextual variable (e.g., the neighbourhood) is represented by a Gaussian distribution with mean $\mu_{HC}$ equal to zero and variance $\sigma_{HC}^2$, from which a value HC is sampled. Sensory evidence about HC is provided by HO, which is sampled from a Gaussian distribution with mean HC and variance $\sigma_{HO}^2$. A lower-level contextual variable (e.g., the restaurant) is represented by a (Gaussian) distribution with mean HC and variance $\sigma_{LC}^2$, from which a value LC is sampled. Sensory evidence about LC is provided and represented by LO which is sampled from a Gaussian distribution with mean LC and variance $\sigma_{LO}^2$. A reward option is obtained and sampled from a Gaussian distribution with mean LC and variance $\sigma_R^2$. We propose that agents infer the posterior distribution P(LC|HO,LO,R) sequentially by estimating, in the order, P(HC|HO), P(LC|HO), P(LC|HO,LO), and P(LC|HO,LO,R). This produces an equation for incentive value with the following form (see Appendix for derivation):

$$V(R) = K(R - \tau_{LO}LO - \tau_{HO}HO) \tag{7}$$

Three normalization effects are implicit here. The first ($\tau_{LO}LO$) is a subtractive normalization proportional to the value LO observed at the low contextual level. A second one ($\tau_{HO}HO$) is a subtractive normalization proportional to the value HO observed at the high contextual level. The terms $\tau$ represent precision-dependent weights and describe the relative precision of the low-level ($\tau_{LO}$) and high-level ($\tau_{HO}$) cues. Finally, a third factor (K) implements divisive normalization and depends on a gain term which includes reward variance (see Appendix).

In summary, this Bayesian formulation outlines a principled theoretical explanation for how we contextualise rewards based on prior expectation and uncertainty with potential deep hierarchical structure. The key role of uncertainty is reflected in the precision-weighting of the prediction errors (e.g., outcome or reward prediction errors).

The proposal advanced here has some similarities with classical theories of value, such as Expected Utility theory (*von Neumann and Morgenstern, 1944*) and Prospect theory (*Kahneman and Tversky, 1979*). For example, there are convergences between the influence of the average reward in BCV and the impact of wealth on marginal utility as postulated in Expected Utility theory. Similarities exist also between the role of the average reward in BCV and the status quo notion in Prospect theory, which distinguishes between loss and profit. Indeed, in BCV profits can be conceived in terms of values larger than the expected reward and losses as values smaller than the expected reward.

We see a more direct link between BCV and recent economic models which postulate that incentive value is adapted to the statistics of the expected reward distribution (Bushong et al., unpublished; *Kőszegi and Rabin, 2006*; *Kőszegi and Szeidl, 2013*), which in turn depends on prior experience within an environment (*Stewart et al., 2006*; *Stewart, 2009*). These theories can be broadly classified into those based on subtractive normalization, which assume that incentive value corresponds to the reward minus a reference value (*Kőszegi and Rabin, 2006*), and those based on divisive normalization, prescribing that incentive value corresponds to the reward divided (or multiplied; *Kőszegi and Szeidl, 2013*) by either the expected reward (*Louie et al., 2014*, *2015*) or the range of an expected distribution of rewards (Bushong et al., unpublished; *Kőszegi and Szeidl, 2013*).

BCV differs in important ways from previous theories in its attempt to derive contextual normalization from normative assumptions of Bayesian statistics. This approach conceives incentive value as precision-weighted prediction error and implies two forms of contextual adaptation (*Figure 2*). First, as in some previous theories (*Kőszegi and Rabin, 2006*), subtractive normalization emerges as the expected reward is subtracted from the actual reward. Second, the gain term implements divisive normalization, an aspect similar to a recent model in which the range of the reward distribution (which is analogous – though not identical - to the gain term) divides the reward (Bushong et al., unpublished). These predictions are specific and distinguish BCV from other models. For instance, BCV predicts that divisive normalization derives from the gain term (i.e., reward variance) and not from the expected reward (*Louie et al., 2014*, *2015*), and that the reward variance divides – and not multiplies (*Kőszegi and Szeidl, 2013*) – the prediction error. Importantly, these predictions are not *ad hoc* but derive necessarily from Bayesian assumptions.

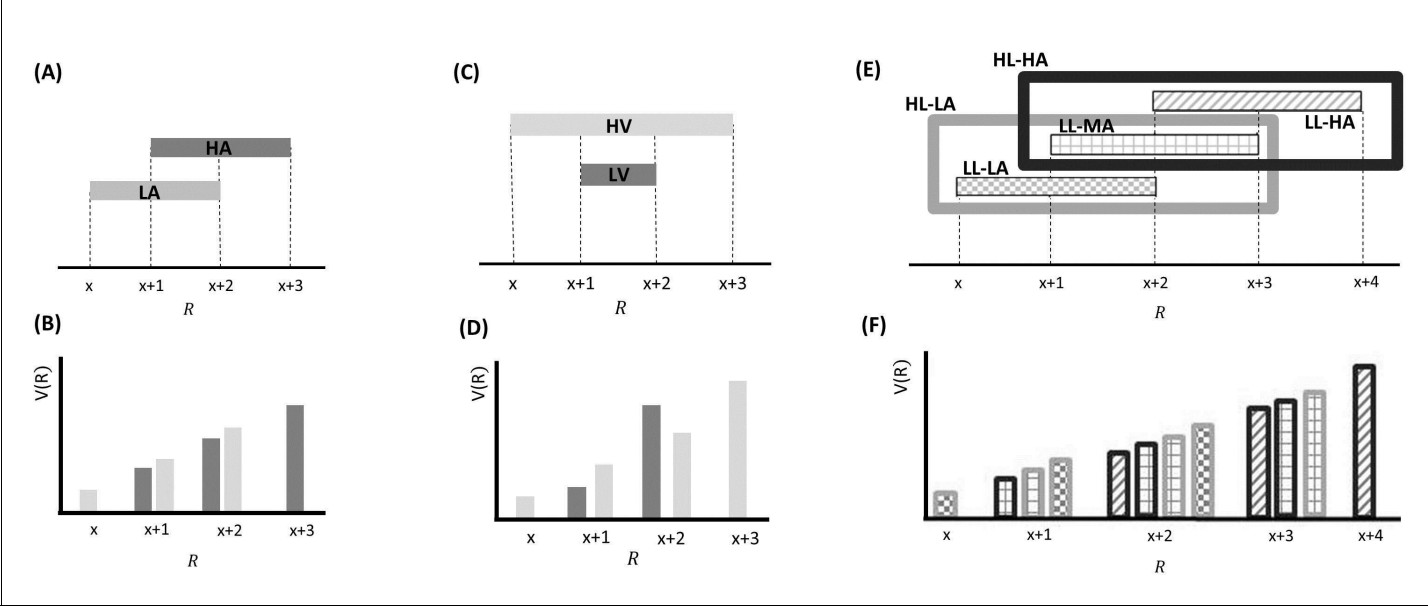

**Figure 2.** Effects predicted by BCV on the incentive value (V(R)) as a function of reward (R) and contexts associated with specific distributions of rewards presented sequentially over trials arranged in blocks. (**A**) Example with a single hierarchical level where two contexts have different average reward. In blocks associated with a low-average context (LA; in lighter grey), the possible rewards are x, x+1 and x+2; in blocks associated with a high-average context (HA; in darker grey), the possible rewards are x+1, x+2 and x+3. (**B**) BCV prediction of the incentive value attributed to rewards depending on these contexts. Larger values are predicted in the LA compared to the HA for amounts common to both contexts. (**C**) Effects predicted by BCV dependent on contexts with different reward variance. In blocks associated with a high-variance context (HV; in lighter grey), the possible rewards are x, x+1 x+2 and x+3; in blocks associated with a low-variance context (LV; in grey), the possible rewards are x+1 and x+2. (**D**) BCV prediction of the incentive value attributed to rewards depending on these contexts. Considering rewards common to both contexts, BCV predicts a higher incentive value for x+1 in the high-variance context and for x+2 in the low-variance context. (**E**) Example with two hierarchical levels (low-level (LL) contexts, represented by patterns of bars, and high-level (HL) contexts, represented by frames). Blocks associated with HL contexts comprise several sub-blocks associated with LL contexts having specific average reward. In the HL context with low-value (HL-LA; light frame), a LL context with low value (LL-LA, where rewards are x, x+1 and x+2) and a LL context with a medium value (LL-MA, where rewards are x+1, x+2 and x+3) alternate. In the HL context with high-value (HL-HA; dark frame), a LL-MA context and a LL context with high value (LL-HA, where rewards are x+2, x+3 and x+4) alternate. (**F**) BCV prediction of the incentive value attributed to rewards depending on these hierarchical contexts. The pattern of bars represents the LL context condition, the outline colour represents the HL context condition. BCV predicts that incentive values derive from integrating both hierarchical levels, with larger values emerging when the average reward is lower at both context levels.

## Experiment one

Data from conditions where BCV is applicable, namely those involving a single attribute and where context depends on past options (and not simultaneously presented options), are relatively scarce. Here, empirical evidence has shown a *subtractive* normalization, whereby incentive values are rescaled to the expected reward (*Kőszegi and Rabin, 2006*; *Ludvig et al., 2013*; *Rigoli et al., 2016a*, *2016b*). In addition, there is an absence of evidence for a *divisive* normalization exerted by the expected reward (i.e., where values are *divided* by the expected reward; *Rigoli et al., 2016a*, *2016b*). Both findings are consistent with BCV. However, another key prediction of BCV relies on a divisive normalization dependent on reward variance (*Figure 2C–D*), though it remains unknown whether such variance-dependent normalization actually occurs. Here, we present data from a behavioural experiment where we investigate this very question.

Participants performed a computer-based decision-making task (*Figure 3*) in which a monetary amount, changing trial by trial, was presented in the centre of the screen and participants had to choose whether to accept half of it for sure or select a 50–50 gamble between the full monetary amount and zero, a scenario where the sure option and gamble always carry the same expected value (EV). The task was organized in blocks, each associated with one of two contexts which determined the possible EVs associated with the block. These EVs were £3 and £4 for the low-variance context, and £2, £3, £4 and £5 for the high-variance context. Note that average choice EV was equal

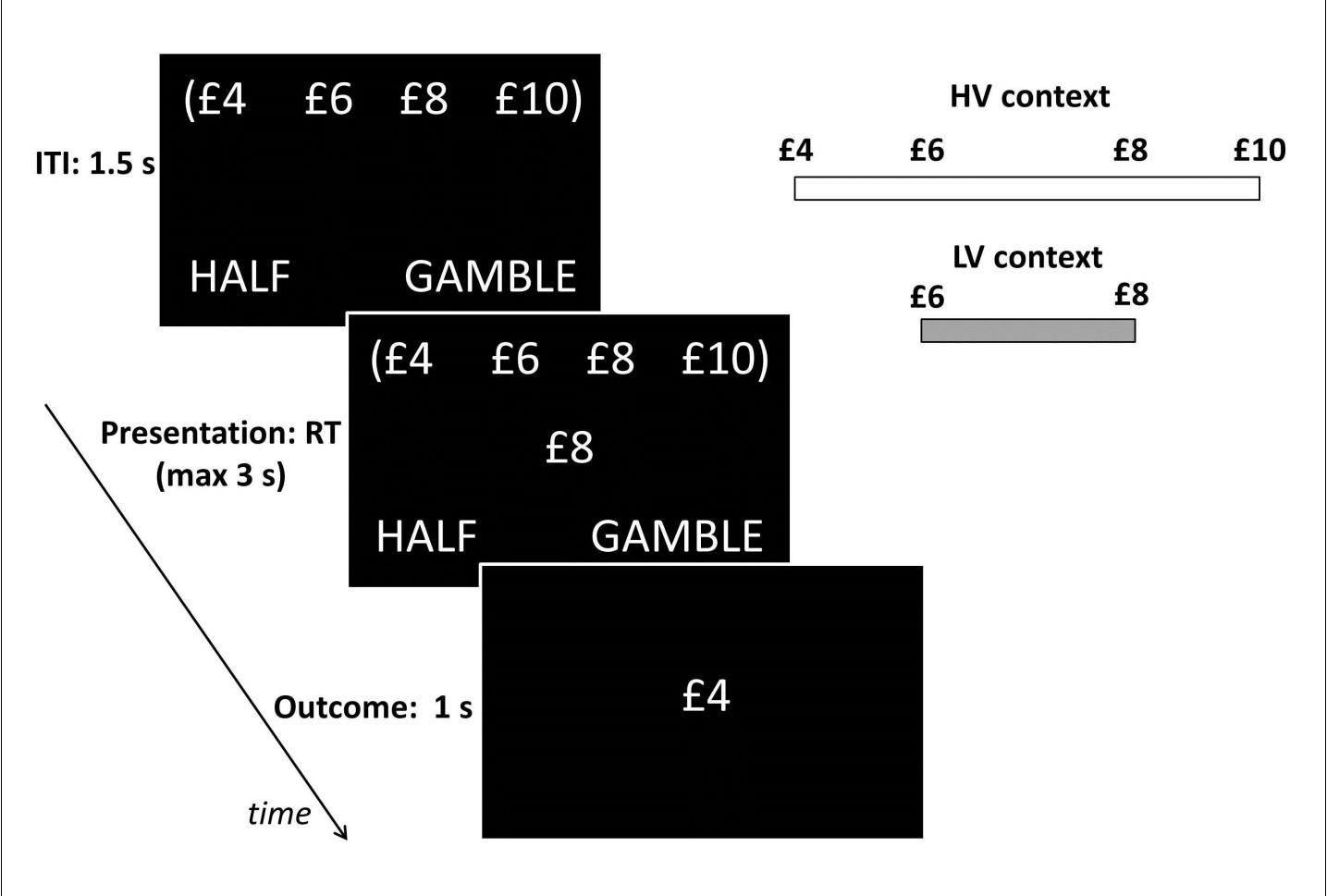

**Figure 3.** Behavioural paradigm. On each trial, participants were presented a monetary amount (£8 in this example) in the middle of the screen and had to choose between half of the amount for sure (pressing a left button) and a gamble (pressing a right button) associated with either the full amount or a zero outcome, each with an equal chance. The outcome was shown right after response for one second. During the intertrial interval (ITI), the possible amounts of the next trial were shown on the top of the screen in brackets. The same amounts were presented throughout blocks associated either with a low-variance (LV) context, including £6 and £8 amounts, or a high-variance (HV) context, including £4, £6, £8 and £10 amounts.

across contexts (i.e., £3.5). Contexts were cued by the associated EVs displayed on the top of the screen in brackets. To ensure incentive compatibility, at the end of the experiment one single outcome was randomly selected and paid out to participants.

In the analyses we focused on choices common to both contexts, namely involving £3 and £4 EV. p<0.05 was used as significance criterion and trials with reaction times slower than 3 s (and faster than 400ms) excluded. For these choices, the average gambling probability did not differ from 50% (mean = 49; SD = 23; t(35) = −0.31, p<0.76). In addition, gambling probability was equivalent when comparing £4 and £3 choices (t(35) = −0.81, p =0.43). Across individuals, there was no correlation between the average gambling probability for £4 and £3 and the difference in gambling probability between these two EVs (*Figure 4C–D;* r(36) = −0.06, p =0.75). The latter result replicates previous findings (*Rigoli et al., 2016a*, *2016b*) and supports the idea of a differentiation between an average gambling propensity and a preference to gamble with large or small EV as determinant of risk choice.

We investigated key predictions of BCV regarding a divisive normalization (or precision weighting) effect exerted by context, which is captured by the gain term $\frac{\sigma_C^2}{\sigma_C^2 + \sigma_R^2}$ in *equation 2*. We can formalize our context manipulation by varying the reward variance $\sigma_R^2$, which is larger in the high-variance context, implying a smaller gain term. Given that £4 and £3 are larger and smaller than the

context average (i.e., £3.5) respectively, we predicted that (at option presentation) they induce a positive and negative prediction error respectively. Because of the gain term, BCV predicts (*Figure 2C–D*) that £4 is attributed a larger (i.e., more positive) incentive value in the low-variance context while £3 is attributed larger (i.e., less negative) incentive value in the high-variance context. Note that in our task there are two types of variance. The first refers to the variance of possible outcomes of the gamble (which is perfectly correlated with the EV of options, as in *Rigoli et al., 2016*), and is not the focus of our study. The second refers to variance *across options* (i.e., the variance characterizing the distribution of successive options), which is what we experimentally manipulate and investigate. In the model, this is reflected in and affects the gain term in *equation 2*.

First, we tested our central predictions by analysing raw choice data. Though we observed no overall difference in gambling across participants for £4 and £3 EV (see above), participants could be differentiated based on those who gambled more with £4 or £3 EV. In line with previous observations (*Rigoli et al., 2016a*, *2016b*), we predicted that the impact of context on gambling depended on a subject-specific propensity to gamble more with large or small choice EVs; in other words on whether a participant prefers to gamble for £3 compared to £4 EV or vice versa. Combining this prediction with BCV predictions (*Figure 2C–D*), participants who risked more with increasing EVs would be expected to gamble more for £4 and less for £3 in the low-variance context (when £4 and £3 would be attributed larger and smaller incentive value respectively) compared to the high-variance context. On the contrary, participants who risked more with decreasing EVs would be expected to gamble less for £4 and more for £3 in the low-variance context compared to the high-variance context. To examine these predictions we tested for an interaction, corresponding to the differential gambling percentage across contexts (low-variance minus high-variance context) for £4 choices minus the differential gambling percentage across contexts (low-variance minus high-variance context) for £3 choices. Across participants, the interaction term did not differ from zero (t(35) = -0.43, p =0.67) but, consistent with our hypothesis, it showed a significant correlation with the gambling probability for £4 minus £3 choices (*Figure 4A,B,E,F;* r(36) = 0.45, p = 0.005). Note that this result remains significant when using a Spearman correlation, which is less affected by outliers (rho = 0.51, p = 0.002).

Next, we adopted a model-based approach to assess whether BCV explains choice data. Following *equation 2*, if the option EV is R, then its associated incentive value will be:

$$V(R) = \tau^{\varepsilon}(R - \mu_C) \tag{8}$$

Where $\varepsilon$ is an indicator of the high ($\varepsilon = 1$) or low-variance context ($\varepsilon = 0$), $\mu_C$ indicates the contextual mean (and is equal to £3.5 for both contexts), and $\tau$ is a free parameter (bounded within the 0.1–10 range) which implements a gain term and captures divisive normalization of reward. To connect value adaptation to choice, we used a logistic regression model of gambling where the probability of choosing the gamble does not depend on the objective option EV, but on the associated incentive value (i.e., transformed by contextual normalization):

$$P(gambling) = \sigma(\alpha V(R) + \rho) = 1/(1 + \exp(-\alpha V(R) - \rho)). \tag{9}$$

Where $\alpha$ is a value-related parameter which determines whether gambling increases ($\alpha > 0$) or decreases ($\alpha < 0$) with larger incentive value and $\rho$ represents a gambling bias parameter.

We used likelihood ratio tests (see Materials and methods) to compare this model with simpler (i.e., reduced) models, where one or more parameters were set to zero. Model comparison favoured the full model (comparison with: random model: $\chi^2(108) = 3268$, p<0.001; model with $\alpha$: = 2502, p<0.001; model with $\rho$: $\chi^2(72) = 1526$, p<0.001; model with $\alpha$ and $\rho$: (36) = 704, p<0.001). In addition, the model predicts that the free parameter $\tau$ is smaller than one or log($\tau$) is less than zero. Consistent with BCV and variance-dependent normalization, the mean of log($\tau$) was significantly smaller than zero (t(35) = $-2.81$, p = 0.008).

We next compared the full model with an alternative model without a subtractive normalization component (as postulated by BCV, where the expected reward is subtracted to the actual reward); namely, where the incentive value was equal to $V(R) = \tau^{\varepsilon}R$. The latter model derives from previous accounts of context-sensitive value (Bushong et al., unpublished; *Kőszegi and Szeidl, 2013*). We compared the negative log-likelihood of the two models (given that they had an equal number of free parameters) and found a smaller score for the model with the subtractive component (difference

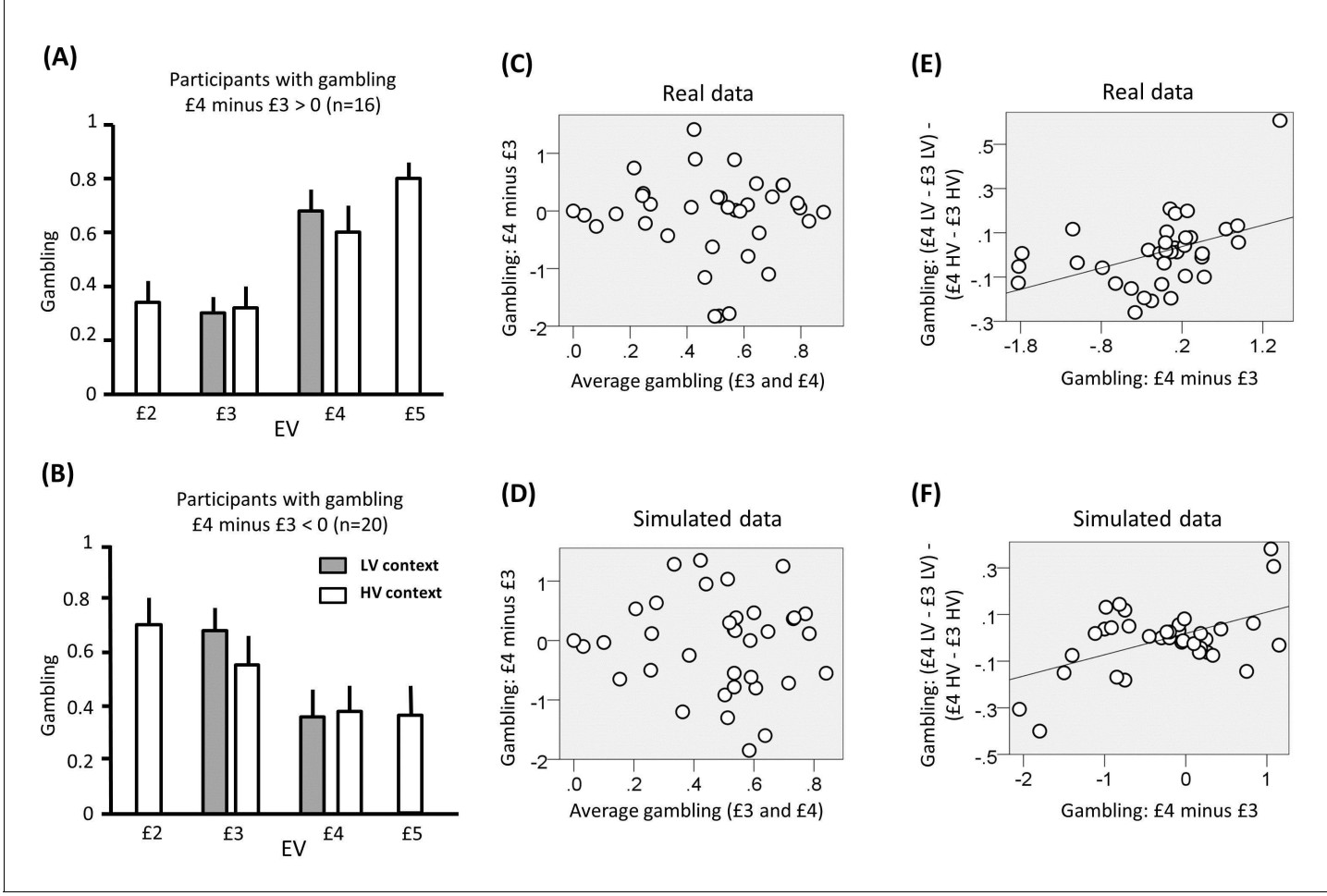

**Figure 4.** Results of experiment one (data used for the analyses reported here are provided in *Source code 1*). (A) Gambling proportion for different EVs and contexts for participants who gambled more for £4 compared to £3 choices. (B) Gambling proportion for different EVs and context for participants who gambled less for £4 compared to £3 choices. (C) Relationship between average gambling proportion for £3 and £4 EV choices and gambling proportion for £4 minus £3 choices. Data showed no correlation (r(36) = -0.06, p = 0.75). (D) The same relationship is reported for data simulated with the computational model of choice behaviour and the parameters estimated from choice behaviour (r(36) = −0.03, p = 0.87). (E) Relationship between (i) the gambling proportion for £4 minus £3 choices and (ii) the interaction term reflecting the gambling proportion for £4 minus £3 choices in the low-variance context compared to the gambling proportion for £4 minus £3 choices in the high-variance context. Data showed a positive correlation (r(36) = 0.45, p = 0.005). Note that this result remains significant also using Spearman correlation, which is less affected by outliers (rho = 0.51, p = 0.002). (F) The same relationship is reported for data simulated with the computational model of choice behaviour and the parameters estimated from choice behaviour (r(36) = 0.48, p = 0.003; the plot reports an example taken from the 100 simulations).

The following figure supplement is available for figure 4:

**Figure supplement 1.** Experiment one: distribution of parameters estimated from choice data with the full model.

in log-likelihood: 254). We tested this difference by performing a Chi-square test with one degree of freedom (treating the model predicted by BCV as having an additional parameter). This test was significant ($\chi^2(1)$ = 508, p<0.001), meaning that the model implementing both subtractive and divisive normalization (derived from BCV and described by *equation 8*) fits the data better.

Finally, we used the (selected) model and subject-specific parameter estimates of the last analysis to generate simulated choice behaviour and performed behavioural analyses on the ensuing data (data were simulated 100 times and the average statistics are reported). Consistent with real data, the full model replicated the lack of correlation between average gambling (for £4 and £3 choices) and the difference in gambling for £4 and £3 choices (average r(36) = −0.06, p = 0.73), while a correlation emerged when data were simulated using a model without the gambling bias parameter ρ

(average r(36) = 0.51, p = 0.001). Moreover, and again consistent with empirical data, the full model replicated the correlation between gambling for £4 minus £3 choices and the context-EV interaction effect term (average r(36) = 0.48, p = 0.003), a result not obtained when data were simulated using a model without the value-function parameter α (average r(36) = 0.01, p = 0.95) or without the context parameter τ (average r(36) = 0.09, p = 0.60).

Collectively, these analyses validate the proposal of a divisive normalization component dependent on a gain term (in turn dependent on reward variance) consistent with BCV.

## Experiment two

Another key prediction of BCV is that the generative model can reflect contexts organized hierarchically and that incentive value and choice are adapted to contextual information available at different hierarchical levels (*Figure 2E–F*). So far, there has been a focus on non-hierarchical settings (*Kőszegi and Rabin, 2006*; *Ludvig et al., 2013*; *Rigoli et al., 2016a*, *2016b*), and therefore whether adaptation combines influence from context at multiple hierarchical levels remains unknown. Here, we present data from a behavioural experiment where we investigated this question.

Participants played a computer-based task (*Figure 5*) where, on each trial, two rectangles representing two decks of cards appeared. Each card was associated with a monetary reward, and the average card reward for each deck was displayed in brackets on the deck. The decks were coloured; one in grey and the other in blue. A card was pseudo-randomly drawn from the blue deck and the corresponding monetary reward was presented in the middle of the screen. Participants had to choose between half of the monetary reward for sure and a gamble between the full reward and a zero outcome, each with 50% chance. Note that, as in experiment one, the two options carry the same EV. After making a choice, the outcome was then shown. To ensure incentive compatibility, at the end of the experiment, one outcome was randomly selected and paid out to participants.

The deck selected by the computer alternated pseudo-randomly over blocks. In addition, two sets of decks alternated over longer blocks in a pseudo-random way. The first deck-set (low-value deck-set) comprised decks associated with an average of £5 and £7, the second (high-value deck-set) comprised decks associated with an average of £7 and £9. The cards of the £5 deck were associated with £3, £5 and £7; the cards of the £7 deck were associated with £5, £7 and £9; the cards of the £9 deck were associated with £7, £9 and £11. The aim of this experimental paradigm was to manipulate context at two hierarchical levels, a low-level associated with decks, and a high-level associated with deck-sets. Note that the rewards overlapped between contexts at both levels; namely, across decks and deck-sets. In relation to decks, £5 and £7 cards were common to both decks in the low-value deck-set, and £7 and £9 cards were common to both decks in the high-value deck-set. If this level of context exerted an influence, it should elicit changes in choice consistent with BCV when comparing choices based upon the same reward across decks. In relation to deck-sets, the deck associated with £7 average was present in both sets. If the deck-set level exerted an influence, this would elicit changes in choice consistent with context sensitive values during the presentation of the £7 deck.

The average gambling percentage did not differ from fifty percent across subjects (mean = 54; SD = 16; t(31) = 1.48, p = 0.12). P<0.05 was used as significance criterion and trials with reaction times slower than 3 s (and faster than 400 ms) excluded. We assessed the impact of option EV on choice using a logistic regression model, where EV was included as regressor. The associated regression coefficient was not significantly different from zero across participants (t(31) = -1.67, p = 0.11). We then investigated the relationship between the average propensity to gamble and the effect of EV on gambling but did not find any correlation (Figure 7A; r(32) = 0.23, p = 0.21). This result again replicates previous studies using a similar paradigm (*Rigoli et al., 2016a*, *2016b*), and highlights two different determinants of risk attitude; one linked with a baseline gambling propensity and the other linked with a preference to gamble with large or small reward amounts.

Though across participants, we observed no overall effect of option EV on gambling, participants could be differentiated based on those who showed a positive or negative effect of option EV on gambling. Based on previous findings (*Rigoli et al., 2016a*, *2016b*), we hypothesized that context sensitive value would predispose participants who preferred to gamble for large EVs to gamble more when EVs were larger relative to contextual expectations, namely in lower value contexts (*Figure 6*). Similarly, we expected participants who preferred to gamble for small EVs to gamble more

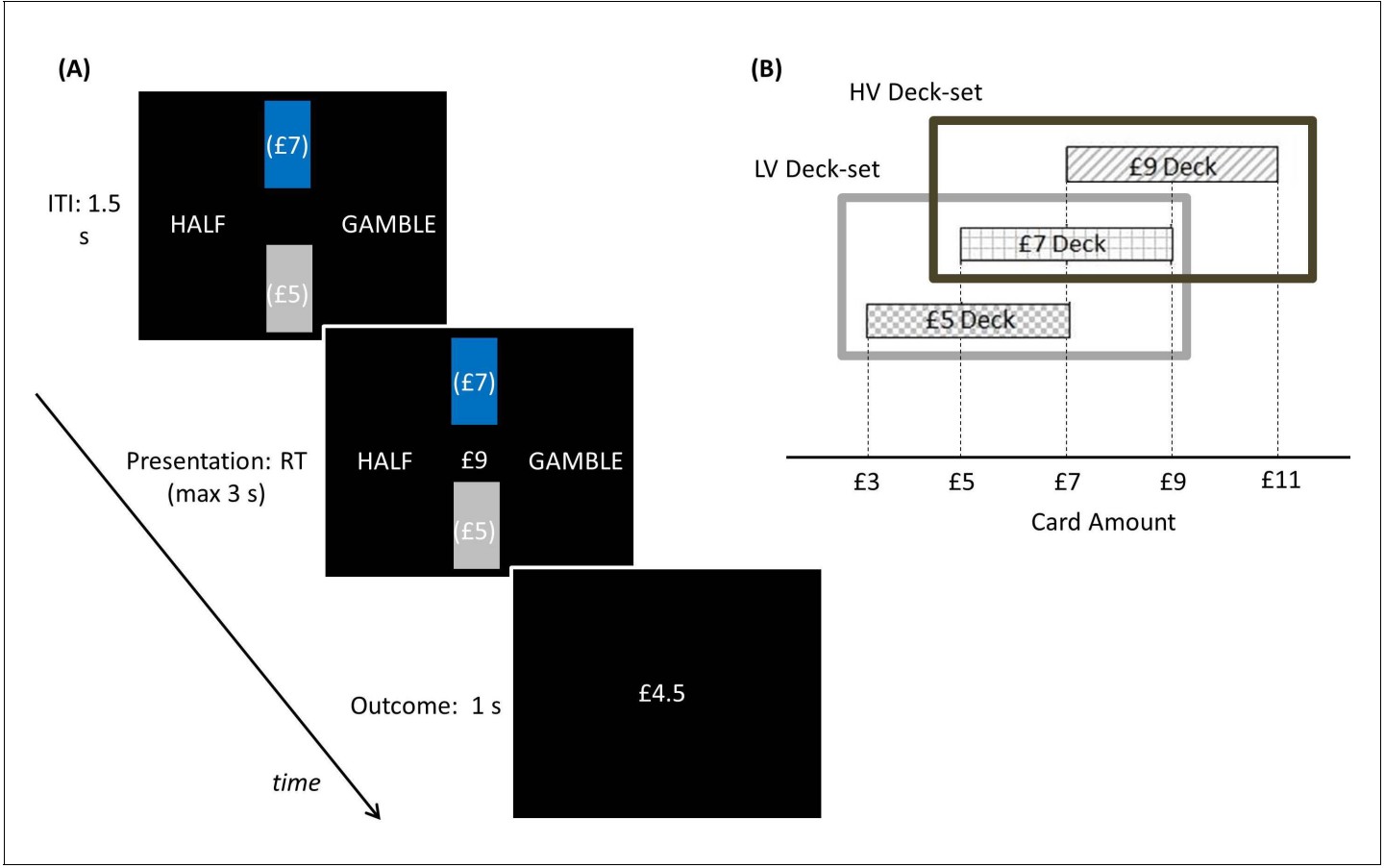

**Figure 5.** Experimental paradigm: participants played a computer-based task where, on each trial, two rectangles representing two decks of cards appeared. (A) Each card was associated with a monetary gain, and the average monetary gain of each deck was displayed in brackets on the deck. The decks were coloured one in blue and the other in grey, indicating the selected and unselected deck respectively. Among these two decks shown on the screen, the selected deck (coloured in blue) alternated pseudo-randomly over blocks (each including 5 trials). In addition, two sets of decks alternated over longer blocks (20 trials) in a pseudo-random way. After decks were shown during an inter-trial-interval of 1.5 s, a card was pseudo-randomly drawn from the blue deck and the corresponding monetary amount was presented in the middle of the screen. Participants had to choose between half of the monetary amount for sure (pressing a left button) and a gamble between the full amount and a zero outcome (pressing a right button), each with a 50% chance. After choosing, the choice outcome appeared for one second. At the end of the experiment, one outcome was randomly selected and paid to participants. (B) Schematic of how contexts are organized in this paradigm. The selected deck alternated pseudo-randomly over blocks. In addition, two sets of decks alternated over longer blocks in a pseudo-random way. The low-value deck-set (LV Deck-set; light grey frame) comprised decks associated with £5 and £7 on average; the high-value deck-set (HV Deck-set; dark grey frame) comprised decks associated with £7 and £9 on average. The cards of the £5 deck could be associated with £3, £5 and £7; the cards of the £7 deck could be associated with £5, £7 and £9; the cards of the £9 deck could be associated with £7, £9 and £11.

when EVs were smaller relative to contextual expectations, namely, in higher value contexts. We investigated this hypothesis both at the level of decks and deck-sets (*Figure 6*).

At the level of decks, we computed – for each deck-set – the difference in gambling between lower and higher value decks for rewards common to both decks (corresponding to £5 and £7 in the low-value deck-set, and to £7 and £9 in the high-value deck-set). The mean of these two differences correlated across subjects with the effect of EV on gambling (i.e., the associated regression coefficient of the logistic regression model; *Figure 7C*; r(32) = 0.55, p = 0.001), consistent with a contextualisation of reward by decks at the lower contextual level. At the level of deck-sets, we computed the difference in gambling between the low and high value deck-set for the £7 deck (common to both deck-sets). This difference correlated across subjects with the effect of EV on gambling (i.e., the associated regression coefficient of the logistic regression model; *Figure 7E*; r(32) = 0.42, p = 0.018), consistent with a context sensitive value effect of the higher contextual level.

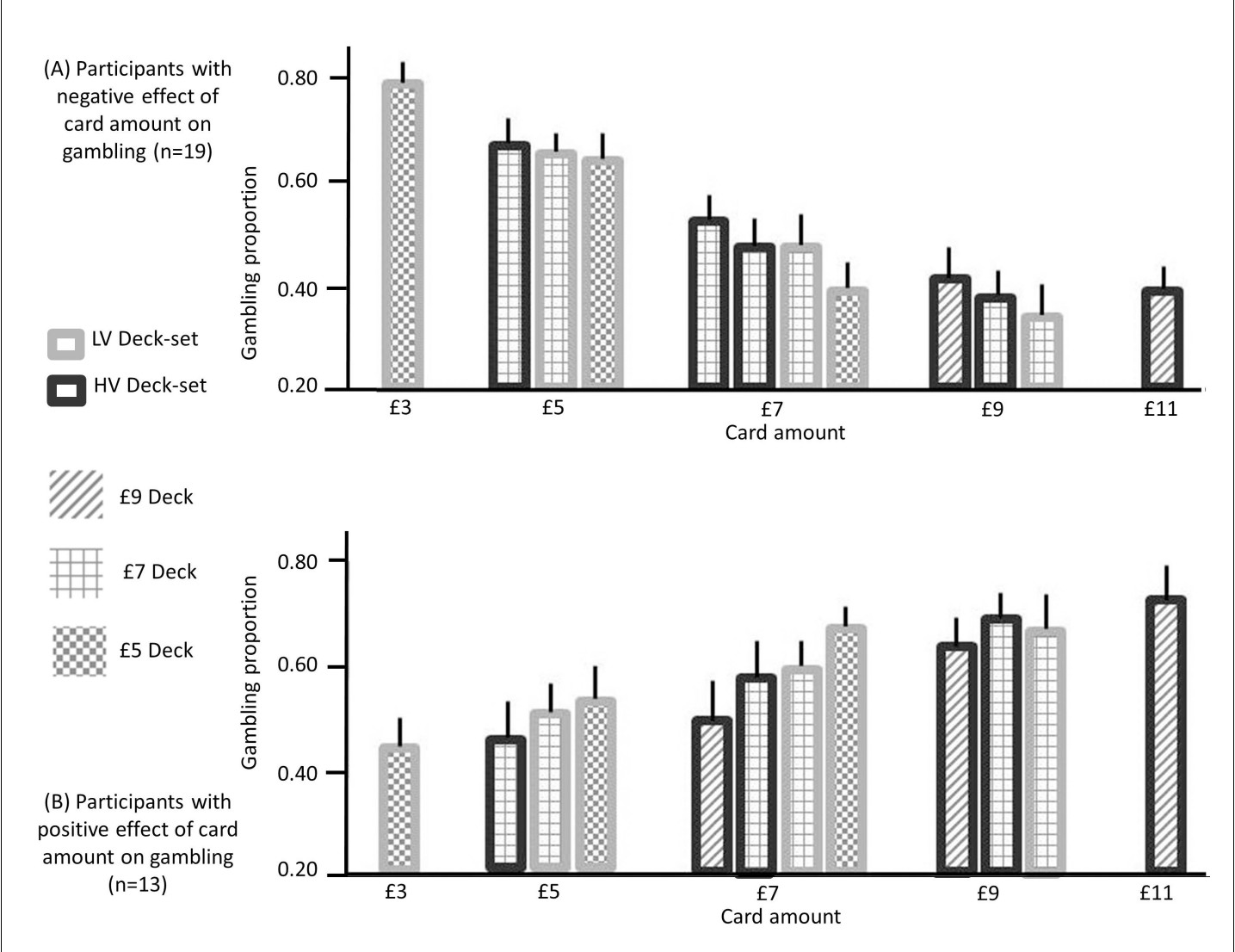

**Figure 6.** Gambling proportion for different card amounts and different context conditions (data used for this figure are provided in *Source code 2*), separately for (**A**) participants showing a negative effect of card monetary amount (i.e., the slope of the logistic regression model with card amount as a predictor) on gambling (n = 19) and (**B**) participants showing a positive effect of the card amount on gambling (n = 13). When considering amounts that are common to multiple contexts, these data show that (i) for the first group of participants, gambling decreases as the context condition is characterized by lower expectations (after integrating both contexts) and (ii) for the second group of participants, gambling increases when the context is characterized by lower expectations, after integrating both contexts (except for the £9 amount). These data are consistent with our hypotheses; namely (i) with predictions arising from BCV (*Figure 2D*) which implies subtractive normalization of incentive value at both hierarchical levels and (ii) with the prediction (derived from previous observations; *Rigoli et al., 2016a*, *2016b*) that the influence of incentive value on gambling proportion depends on the individual preference to gamble with large or small card amounts.

We next investigated whether BCV explains the context effects implicit in these results. Since, in our task, contexts are organized hierarchically (i.e., decks and deck-sets are associated with high and low levels – in the sense that the set determines the possible decks), we refer to the generative model shown in *Figure 1C*, where incentive value is described by *equation 7*. Recall that there are three normalization terms under this model (see *equation 7*): first a subtractive term ($\tau_{Lo}LO$) proportional to the value LO observed at the low contextual level, second a subtractive term ($\tau_{Ho}HO$) proportional to the value HO observed at the high contextual level, and third a divisive factor (K) dependent on precision. In our task, the average contextual reward was manipulated, enabling us to examine the subtractive terms outlined in *equation 7*. We did this by treating the contextual

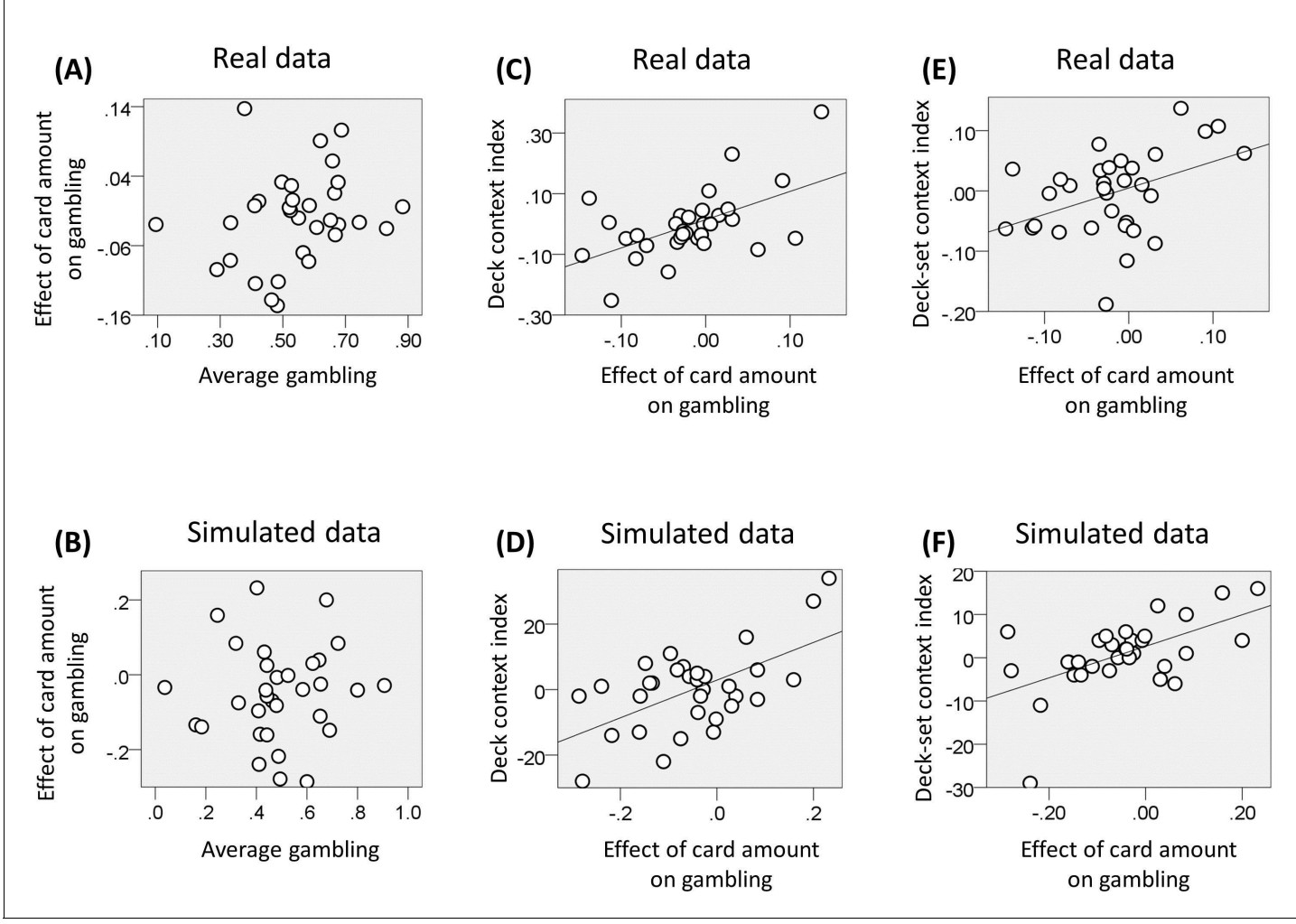

**Figure 7.** Results of experiment two (data used for the analyses reported here are provided in *Source code 2*). (A) Relationship between the individual gambling percentage and the effect of monetary amount on gambling proportion (i.e., the associated regression coefficient of the logistic regression model; r(32) = 0.23, p = 0.21, non-significant). (B) The same relationship is reported for data simulated with the computational model of choice behaviour and the parameters estimated from choice behaviour (r(32) = 0.06, p = 0.76, non-significant; the plot reports an example taken from the 100 simulations). (C) We computed for each deck-set the difference in gambling between lower and higher value decks for amounts common to both decks; corresponding to £5 and £7 in the low-value deck-set, and to £7 and £9 in the high-value deck-set. The relationship between the mean of these two differences and the effect of card amount on gambling is reported (r(32) = 0.55, p = 0.001). (D) The same relationship is reported for data simulated with the computational model of choice behaviour and the parameters estimated from choice behaviour (r(32) = 0.58, p<0.001; the plot reports an example taken from the 100 simulations). (E) Relationship between the difference in gambling between the low and high value deck-set for the £7 deck (common to both deck-sets) and the effect of card amount on gambling (r(32) = 0.42, p = 0.018). (F) The same relationship is reported for data simulated with the computational model of choice behaviour and the parameters estimated from choice behaviour (r(32) = 0.57, p = 0.03; the plot reports an example taken from the 100 simulations).

The following figure supplement is available for figure 7:

**Figure supplement 1.** Experiment two: distribution of parameters estimated from choice data with the full model.

averages as observations of the true underlying contexts of the task (in *equation 7*, HO and LO would be associated with deck-set and deck respectively). However, since reward variance (which enters the divisive normalization factor K) was not manipulated, this task is not suitable to quantify an effect of precision-weighting – and therefore we omitted this factor from the model of our empirical data. Specifically, based on *equation 7* and omitting K, if the option EV is R, then its associated incentive value will be:

$$V(R) = R - \chi_{LO}\tau_{LO} - \chi_{HO}\tau_{HO} \tag{10}$$

Where $\chi_{LO}$ indicates the average option EV for the deck (for £9 deck: $\chi_{LO}$ = 4.5; for £7 deck: $\chi_{LO}$ = 3.5; for £5 deck: $\chi_{LO}$ = 2.5), $\chi_{HO}$ indicates the average option EV for the deck-set (high-value deck-set: $\chi_{HO}$ = 4; low-value deck-set: $\chi_{HO}$ = 3) $\tau_{LO}$ is a free parameter that mediates contextual effects at the deck level, and $\tau_{HO}$ is a free parameter that mediates contextual effects at the deck-set level (see Materials and methods).

To connect value adaptation to choice, we used a logistic regression model as in *equation 9* where $\alpha$ is a value-related parameter which determines whether gambling increases ($\alpha > 0$) or decreases ($\alpha < 0$) with larger incentive value and $\rho$ represents a gambling bias parameter. We used a likelihood ratio test to compare this model with simpler (reduced) models where one or more parameters were set to zero. The full model was favoured (comparison with: random model: $\chi^2(128)$ = 2959, p<0.001; model with $\alpha$: $\chi^2(96)$ = 1530, p<0.001; model with $\rho$: $\chi^2(96)$ = 1622, p<0.001; model with $\alpha$ and $\rho$: $\chi^2(64)$ = 262, p<0.001; model with $\alpha$, $\rho$ and $\tau_{LO}$(32) = 134, p<0.001; model with $\alpha$, $\rho$ and $\tau_{HO}$: $\chi^2(32)$ = 58, p<0.001). In addition, consistent with context sensitivity at both hierarchical levels, the context-related free parameters of the full model were significantly larger than zero (*Figure 6*; $\tau_{LO}$: t(31) = 4.55, p<0.001; $\tau_{HO}$ t(31) =2.67, p = 0.012).

We next compared the full model with an alternative model where the context parameters (capturing the influence of the reward expected within a context at multiple hierarchical levels) divided the reward rather than being subtracted from the reward; in other words where the incentive value corresponds to:

$$V(R) = \frac{R}{1 + \chi_{LO}\tau_{LO} + \chi_{HO}\tau_{HO}} \tag{11}$$

The latter model derives from previous accounts of context sensitivity (*Louie et al., 2014*, *2015*). We compared the negative log-likelihood of the two models (given they had an equal number of free parameters) and found a smaller score for the model with subtractive normalization (difference in log-likelihood for divisive minus subtractive model; full models: 39; for models with $\alpha$, $\rho$ and $\tau_{LO}$: 13; for models with $\alpha$, $\rho$ and $\tau_{LH}$: 25). This difference was tested statistically by performing a Chi-square test with one degree of freedom (treating the model predicted by BCV as having an additional parameter). This test was significant (full models: $\chi^2(1)$ = 78, p<0.001; for models with $\alpha$, $\rho$ and $\tau_{LO}$: $\chi^2(1)$ = 26, p<0.001; for models with $\alpha$, $\rho$ and $\tau_{LH}$: $\chi^2(1)$ = 50, p<0.001), meaning that the model implementing subtractive normalization (consistent with BCV) is a better explanation for the data.

Finally, we used the (selected) model and subject-specific parameter estimates of the last analysis to generate simulated choice behaviour and performed behavioural analyses on the ensuing data (data were simulated 100 times and the average statistics are reported). As with real data, our modelling replicated the absence of correlation between average gambling and the effect of reward on gambling (i.e., the slope of the logistic regression; see above) (*Figure 7B*; average r(32) = 0.06, p = 0.76), while a correlation emerged when the data were simulated using a model without the gambling bias parameter $\rho$ (average r(32) = 0.85, p<0.001). Moreover, consistent with empirical data, the full model replicated the correlation between (i) the effect of EV on gambling and the difference in gambling across decks for choices common to both decks of a deck-set (combining both deck-sets; *Figure 7D*; average r(32) = 0.58, p<0.001), (ii) the effect of EV on gambling and the difference across deck-sets in gambling for the £7 deck (common to both deck-sets; *Figure 7F*; average r(32) = 0.57, p=0.001). These correlations were not replicated when data were simulated using a model without the value-related parameter $\alpha$ (first correlation: average r(32) = 0.09, p = 0.62; second correlation: average r(32) = =0.02, p = 0.89). Furthermore, the first correlation was not replicated when using a model without the parameter $\tau_{LO}$ (average r(32) = −0.15, p = 0.41) and the second correlation was not obtained when using a model without the parameter $\tau_{HO}$ (average r(32) = −0.03, p = 0.87).

Collectively, these analyses show subtractive normalization exerted by contextual effects at multiple hierarchical levels consistent with predictions from BCV.

# Discussion

We propose a Bayesian scheme (BCV) as a model of contextual influences on incentive value attribution. BCV is based on Bayesian inference principles and on generative models of reward. Adopting two novel experimental designs, we provide behavioural evidence that supports two key predictions of BCV, namely that value attribution is affected by reward variance (which exerts divisive normalization) and by hierarchically organized contexts.

Our account is motivated by normative principles of Bayesian statistics – and fits within a Bayesian brain hypothesis framework (*Chater et al., 2006*; *Clark, 2013*; *Dayan et al., 1995*; *Ernst, 2006*; *Friston, 2010*). As such, it provides a principled account of decision making under uncertainty. In particular, it accommodates expectation and uncertainty that may have a deep hierarchical structure, as in real world situations. Bayesian schemes are based on a formal and a clear definition of the imperatives that motivate cognitive processes, which are conceived in terms of inference. This allows BCV to establish a link with Bayesian perspectives in other domains of cognitive neuroscience, helping unifying perspectives on brain functioning.

Our proposal is closely linked to the framework of planning as inference and active inference (*Botvinick and Toussaint, 2012*; *Friston et al., 2013*, *2015*; *Pezzulo and Rigoli, 2011*; *Pezzulo et al., 2015*; *Solway and Botvinick, 2012*). This recasts decision-making and planning – usually understood in terms of value or utility maximization – as a form Bayesian inference, and hence can provide a unifying inferential account of perception and action. Hierarchical implementations of active inference schemes have been proposed previously, and the notion of hierarchically-organized contexts fits comfortably within these schemes (*Pezzulo et al., 2015*). BCV extends this framework by focusing on the determinants of incentive value, conceived as precision-weighted prediction error based on (potentially hierarchical) contextual expectations.

BCV postulates that the two fundamental determinants of incentive value are prediction error and precision. A prediction error is determined by the difference between the observed and expected reward which, in BCV, derives from integrating different expectations under contextual uncertainty. Gain depends on (relative) precision or confidence– and ensures that the prediction error is normalised and (Bayes) optimally weighted in relation to uncertainty about both context and reward. In brief, only precise prediction errors have an effect on expectations higher in the hierarchy during Bayesian belief updating. BCV predicts that precision exerts an influence in two ways. First, at the highest hierarchical levels, precision determines the optimal integration of multiple contextual representations–as it mandates that contexts characterized by a high precision (greater reliability) exert more influence on reward expectancy. For instance, if we assume that subjects have very precise beliefs about the low-level context (e.g., the deck), then the effect of the high-level (e.g., the deck set) will disappear. Formally, this is because in the hierarchical model the low-level context constitutes a Markov blanket for the posterior expectation about the reward option (*Bishop, 2006*). In other words, the effect of the high-level context tells us that if subjects are using a hierarchical model, there must be posterior uncertainty about the low-level context. Heuristically, even though they can see which deck they are currently playing with, they still nuance their expectations about this deck based upon the deck-set from which it came. Second, at the lowest hierarchical level, the precision determines the gain assigned to the prediction error and hence is a direct determinant of incentive value. In our first experiment, we show evidence consistent with the latter expression of precision.

The central role attributed to precision-weighted prediction error is consistent with Bayesian models in other domains, and speaks to the idea of common computational principles in the brain. In fact, one central idea of many influential Bayesian proposals is that, when sensory inputs are presented, predictions are explained away and the resulting perception corresponds to (precision-weighted) prediction errors. For instance, predictive coding models are based on evidence that activity in certain brain regions responsible for perception reflects prediction errors and not raw sensory inputs (*Friston, 2005*; *Rao and Ballard, 1999*). Moreover, it has been proposed that our sense of agency depends on explaining away somatosensory predictions associated with motor commands, with unexplained sensations alone (i.e., residuals) attributed to external forces (*Brown et al., 2013*). A similar view characterizes active inference schemes, which assume that action is not steered by stimuli per se but by the (precision-weighted) prediction error elicited by those stimuli, expressing the extent to which they depart from expectation in a meaningful way (*Friston et al., 2013*, *2015*).

Within BCV contextual representations can be hierarchically organized, with high levels characterized by more general conditions. Because reward options are descendants of all hierarchical levels, any context can exert an influence on incentive value, insofar as these levels determine the reward expected in a certain condition. Specifically, BCV can integrate–in a Bayes optimal way–context-independent beliefs about the reward distribution with context-sensitive beliefs unfolding at multiple hierarchical levels. The possibility that subjects use hierarchical generative models is in fact supported by our empirical findings. Our results are consistent with the idea that rewards have larger incentive values when both high and low-level contexts are characterized by reward distributions with a smaller average. This indicates that information about more specific (e.g., the deck) and more general (e.g., the deck-set) contexts are integrated to determine incentive values and choice behaviour.

A hierarchical nesting might explain why contextual effects observed in psychological experiments are usually substantial but not extreme. In other words, it is unlikely that 10 p will be attributed the same value as £100, even when the contextual manipulation may appear to induce an equivalence between the two quantities. This can be explained by contextual effects from the highest hierarchical level (e.g., that reward options have, in general, a prior expectation of zero). This supraordinate level can be conceived as representing a context-independent distribution of rewards that may derive from the overall statistics of our environment (*Stewart et al., 2006*, *2009*) and/or from innate prior beliefs about the distribution of incentives (*Rigoli et al., 2016c*, *2016d*).

The proposal that incentive value corresponds to (precision-weighted) reward prediction error should not be confounded with the idea that it corresponds to the posterior reward expectation. Though both possibilities derive from Bayesian principles, they make opposite predictions about the role of prior reward expectancy. While the posterior reward expectation hypothesis predicts the larger value with larger prior reward expectancy, our data show larger value with smaller prior expectancy, consistent with the prediction error hypothesis presented here (see also *Rigoli et al., 2016a*, *2016b*).

With reference to the three levels of analysis (i.e., computational, algorithmic and implementation) proposed by *Marr (1982)*, BCV speaks to the computational level as it focuses on normative principles (implicit in optimal Bayesian inference) proposed to explain value and choice adaptation. In addition, BCV also has implications for the other levels and there are now several biologically-plausible accounts of how Bayesian inference might be implemented in the brain (e.g., *Doya et al., 2007*; *Friston, 2005*; *Hennequin et al., 2014*; *Knill and Pouget, 2004*), where some accounts consider neuronal circuits (and generative models) characterized by a hierarchical organization (e.g., *Friston, 2005*). BCV fits comfortably within these biologically-plausible accounts. Consistent with BCV are findings that several brain regions show a response to reward that adapts to both expected reward (e.g., in signalling reward prediction error) and reward range (*Bermudez and Schultz, 2010*; *Cox and Kable, 2014*; *Louie et al., 2011*; *Padoa-Schioppa, 2009*; *Padoa-Schioppa and Assad, 2008*; *Park et al., 2012*; *Rigoli et al., 2016a*; *Tobler et al., 2005 Kobayashi et al., 2010*; *Tremblay and Schultz, 1999*). Recent evidence for an association between neural and choice adaptation is also in line with BCV (*Rigoli et al., 2016a*). However, key neurobiological predictions of BCV, including the specific neural mechanisms that realize choice adaptation as well as the implementation of hierarchical generative models of reward, await further investigation.

Theoretical work has indicated that adaptation of neuronal responses is consistent with efficient coding, whereby the signalling of a finite pool of neurons – with a finite dynamic range of responses – can be optimally tuned to the statistics of stimuli in the environment, so as to maximize discriminability among the stimuli (*Carandini and Heeger, 2012*; *Louie et al., 2015*; *Rangel and Clithero, 2012*; *Summerfield and Tsetsos, 2015*.) This idea has now been extended to reward processing. Here proposals diverge as to the prediction of whether adaptive neuronal coding determines either stability or adaptation in choice behaviour (*Louie et al., 2015*; *Padoa-Schioppa and Rustichini, 2014*; *Rangel and Clithero, 2012*; *Summerfield and Tsetsos, 2015*). In addressing this, BCV (similar to other Bayesian inference schemes for continuous variables) implements adaptive coding, because precision-weighted prediction error postulated to be signalled by value-processing neurons is a normalized quantity (*Doya et al., 2007*). Also in line with previous accounts (*Louie et al., 2015*; *Rangel and Clithero, 2012*; *Summerfield and Tsetsos, 2015*), BCV proposes that this normalized signal corresponds to incentive value, and that adaptive coding in the brain should be reflected in a behavioural choice adaptation. In other words, BCV implies that, as well as neural signalling,

behaviour itself is tuned to the statistics of the incentives, so as to maximize discriminability among these incentives.

We highlight shortcomings of the model, though the framework itself may be fruitful in addressing some of these shortcomings. First, our focus is on scenarios where the incentive value depends on contextual information (either represented by cues or by previous rewards) provided before reward delivery. Another form of context effect on incentive value is induced by options that are simultaneously available (*Louie et al., 2013*, *2015*). Further theoretical work is needed to link BCV and this form of influence, though a sequential inferential process (e.g. Bayesian belief updating), similar to the process described here (possibly linked to attention) might be involved in simultaneous contextual effects. Second, our focus has been on conditions where reward is defined by a single attribute (e.g., reward amount). Contextual influences (e.g., the decoy effect) can emerge when multiple dimensions need to be evaluated and integrated, as investigated by multi-attribute theories (*Huber et al., 1982*; *Johnson and Busemeyer, 2005*; *Roe et al., 2011*; *Simonson and Tversky, 1992*; *Soltani et al., 2012*; *Tsetsos et al., 2010*, *2012*, *2016*; *Tversky, 1972*; *Usher and McClelland, 2004*). BCV can in principle be extended to these scenarios, for instance connecting to a body of work on multisensory integration using Bayesian principles (*Ernst, 2006*). This would provide an opportunity to model attentional processes determining an optimal weighting of different attributes based on their importance and reliability. Third, our current formulation assumes that the model parameters are given, while these parameters need to be learned in the first place. Questions about the mechanisms that might underpin learning of generative models adopted for Bayesian inference are still largely open, though substantial contributions exist particularly in the context of structure learning (*Acuna and Schrater, 2009*; *Behrens et al., 2007*; *Collins and Frank, 2013*; *Courville et al., 2006*; *FitzGerald et al., 2014*; *Gershman and Niv, 2010*; *Mathys et al., 2011*).

Here, we have assumed that variables of the generative model are Gaussian. This allows us to present the model in a simple and clear way, as posterior beliefs can be inferred analytically with relatively simple equations as adopted in standard decision-making schemes (*Rescorla and Wagner, 1972*). Though Gaussian assumptions are probably an over-simplification, with appropriate adjustments BCV can be extended to generative models with non-Gaussian variables (*Jazayeri and Shadlen, 2010*). Indeed, the arguments behind BCV can be applied to any variables with an exponential distribution. However, the key idea (tested in our experiments – and here derived from Gaussian assumptions) that reward average and variance elicit subtractive and divisive normalization, respectively, is quite general and can also be applied, for instance, to uniform (and, in general, non-skewed) distributions.

Finally, there are questions related to psychopathology that can be fruitfully formulated in terms of BCV, for example addiction. Consider the consequences of drug misuse, including the development of tolerance (i.e., the need of increased dosages to obtain the same effects as those obtained previously) and the lack of satisfaction when engaging in activities that were pleasurable before the development of addiction. BCV interprets these effects in terms of increases in expected reward (following drug misuse) that decreases the incentive value of rewards, including the drug itself and other motivational stimuli. A similar explanation has been proposed by classical homeostatic theories, where ingestion of the drug is conceived in terms of a means to re-establish a biological set point (e.g., expressed in baseline activity of dopamine neurons), coupled with the fact that the repeated drug misuse raises this set point. BCV formalise and extend the set-point model. First, it relaxes the homeostatic assumption because incentive value depends on a reference point (i.e., the reward average), but does not correspond to distance from a set point as in homeostatic schemes. Indeed, set-point models predict that drug consumption always reduces a negative affect state by re-establishing a set point. Conversely, BCV suggests that drug consumption can decrease a negative state (when the drug-associated outcome is worse than expected) but also induce a positive affect (when the drug-associated outcome is better than expected), a prediction more consistent with empirical evidence (*Robinson and Berridge, 2000*). Second, leveraging a Bayesian framework, BCV assigns a crucial role to reward uncertainty, above and beyond a role assigned to expected reward. For instance, increased uncertainty over prior reward beliefs may boost the magnitude of the (positive) prediction error elicited by drug consumption, hence enhancing individual predisposition to drug addiction.

In summary, we introduce a normative Bayesian model to explain the influence of contexts on incentive values. Key features of this account include an explicit generative model of reward and the

assumption that incentive value corresponds to precision-weighted prediction error. This formulation implies that incentive value is intrinsically context-dependent. We tested key predictions of the model in two human experiments and show choice behaviour consistent with an adapting incentive value based on reward variance and on an average reward expected after integrating contexts at two hierarchical levels, one more general and the other more specific. An important consideration is that expression of context effect, though apparently irrational, can derive from Bayes (optimal) inference. Indeed, if incentive values are (precision-weighted) prediction errors, they are necessarily context-dependent, and this dependency can be described under a Bayes optimal scheme. We argue that this approach could be useful in generating new empirical predictions and in explaining phenomena in psychopathologies characterized by dysfunctional value attribution, such as addiction.

## Materials and methods

### Participants

36 healthy right-handed adults (19 females; 20–40 age range; mean age 26) participated in experiment one, and 32 healthy right-handed adults (18 females, aged 20–40, mean age 27) participated in experiment two. All participants had normal or corrected-to-normal vision. None had a history of head injury, a diagnosis of any neurological or psychiatric condition, or was currently on medication affecting the central nervous system. The first experiment was conducted at the Wellcome Trust Centre for Neuroimaging at the University College London and was approved by the University College of London Research Ethics Committee. The second experiment was conducted at the Institute of Psychiatry, Psychology & Neuroscience at the King's College of London and was approved by the King's College of London Research Ethics Committee. All participants provided written informed consent and were paid for participating.

### Experimental paradigm and procedure

#### Experiment one

Participants performed a computer-based decision-making task lasting approximately 30 min. A monetary amount (referred as trial amount), changing trial by trial, was presented in the centre of the screen and participants had to choose whether to accept half of it for sure (pressing a left button) or select a gamble (pressing a right button). The outcomes of this choice were either zero or the full monetary amount, each with equal probability, ensuring the sure option and gamble always had the same EV. The task was organized in blocks, each associated with one of two contexts which determined the possible EVs associated with the block. These EVs were £3 and £4 for the low-variance context, and £2, £3, £4 and £5 for the high-variance context. Note that average choice EV was equal across contexts (i.e., £3.5). Contexts were cued by the associated EVs displayed on the top of the screen in brackets. Before a new block started, the statement 'New set' appeared for two seconds, followed by the contextual cue for two seconds. Next, the trial amount of the first trial was displayed followed, after the subject made a choice, by the outcome shown for one second. The possible contextual cue remained on the screen during an inter trial interval that lasted one and a half seconds. Participants had three seconds to make their choices; otherwise the statement 'too late' appeared and they received a zero outcome amount.

Eight blocks were presented, alternating between a low and high-variance context with the order counterbalanced across subjects. The former blocks comprised 40 trials each and the latter blocks comprised 80 trials, in such a way that the EVs common to both contexts (i.e., £3 and £4) were shown an equal amount of time in the two contexts. The order of trial amounts and outcomes were pseudo-randomized. At the end of the experiment, one outcome was randomly selected among those received and added to an initial participation payment of £5. Before the task, participants were fully instructed both on task contingencies and payment rules.

#### Experiment two

Participants played a computer-based task lasting approximately 40 min. On each of the 480 trials, two rectangles representing two decks of cards appeared, one on the top and the other on the bottom of the screen. Each card was associated with a monetary amount, and the average amount of each deck was displayed in brackets upon the deck. The decks were coloured: one in gray and the

other in blue and were shown for 1.5 s. On each trial a card was pseudo-randomly drawn from the blue deck and the corresponding monetary amount was presented in the middle of the screen. Participants had to choose between half of the monetary amount for sure (pressing a right button) and a gamble between the full amount and a zero outcome (pressing a left button), each with 50% chance. After choosing, the outcome appeared for one second and a new trial started immediately. If no response occurred before three seconds, a statement 'too late' was presented for one second, resulting in a zero outcome.

Among the two decks shown on the screen, the selected deck (coloured in blue) alternated pseudo-randomly over blocks (each including 5 trials). In addition, at some points during the task, the decks were replaced by new decks. Two sets of decks alternated over blocks of 20 trials in a pseudo-random way. The first deck-set (low-value deck-set) comprised decks returning £5 and £7 on average, the second deck-set (high-value deck-set) comprised decks returning £7 and £9 on average. The cards of the £5 deck could be associated with £3, £5 and £7; the cards of the £7 deck could be associated with £5, £7 and £9; the cards of the £9 deck could be associated with £7, £9 and £11. When a new deck was selected (i.e., it was coloured in blue), decks were shown for 2.5 s before the card amount appeared; when a new deck-set appeared, decks were shown for 4.5 s before the card amount appeared. At the end of the experiment, one of the outcomes was randomly selected by the computer, added to an initial payment of £5 and the total amount was paid to participants. The participants were fully instructed about task rules and about the way payment was carried out prior to task performance.

## Behavioural modelling

The free parameters of the models were estimated separately for each subject using *fminsearchbnd* function of the Optimization toolbox in Matlab. Parameters were constrained within the following ranges: −5 and 5 for α (in both experiments), −10 and 10 for ρ (in both experiments), 0.1 and 10 for τ (in experiment one), −1 and 1 for $\tau_{LC}$ and $\tau_{HC}$ (in experiment two). In addition, to minimize the effect of biased outlier estimates, Gaussian priors with mean zero (*Daw, 2011*) were used for estimation of log(τ) (in experiment one) and estimation of $\tau_{LC}$ and $\tau_{HC}$ (in experiment two). Starting values for parameter estimation was zero for all parameters, except for the context parameter τ in experiment one for which it was one. Distributions of estimated parameters are reported in *Figure 4—figure supplements 1* (experiment one) and *Figure 7—figure supplements 1* (experiment two), and show no evidence of outliers (i.e., scores larger or smaller than 3 SD compared to the mean) for any of the parameters.

For each model, the log-likelihood of the choice data given the best fitting parameters (estimated by the method described above) was computed subject by subject and summed across subjects. We compared the full model with nested models, namely where one or more parameters were fixed to zero. To do this, we used the standard approach of the likelihood-ratio test (*Casella and Berger, 2002*; *Daw, 2011*), which allows for a comparison of nested models. This is based on the fact that the difference in negative log-likelihood times two (2*d*) between a nested and a more complex model follows a chi-square distribution, where the number of degrees of freedom is equal to the number of additional parameters of the more complex model. A chi-square test can be performed to estimate the probability that the observed 2*d* is due to chance under the null hypothesis that data are generated by the nested model, allowing acceptance or rejection of the null hypothesis.

## Acknowledgements

This work was supported by the Wellcome Trust (Ray Dolan Senior Investigator Award 098362/Z/12/Z) and the Max Planck Society. The Wellcome Trust Centre for Neuroimaging is supported by core funding from the Wellcome Trust 091593/Z/10/Z. SSS is supported by a consolidator award from the European Research Council. C.M is funded by the NIHR-BRC at South London and Maudsley NHS Foundation Trust and Institute of Psychiatry, Psychology and Neuroscience King's College London via a research studentship.

## Additional information

### Funding

| Funder | Grant reference number | Author |
|---|---|---|
| Wellcome Trust | 098362/Z/12/Z | Francesco Rigoli<br>Karl J Friston<br>Raymond J Dolan |
| European Research Council | | Sukhwinder S Shergill |
| NIHR-BRC | | Cristina Martinelli |

The funders had no role in study design, data collection and interpretation, or the decision to submit the work for publication.

### Author contributions

FR, Conception and design, Acquisition of data, Analysis and interpretation of data, Drafting or revising the article; KJF, RJD, Conception and design, Analysis and interpretation of data, Drafting or revising the article; CM, MS, SSS, Acquisition of data, Drafting or revising the article

### Author ORCIDs

Francesco Rigoli, http://orcid.org/0000-0003-2233-934X
Karl J Friston, http://orcid.org/0000-0001-7984-8909

### Ethics

Human subjects: Experiment one was approved by the University College London Research Ethics Committee. Experiment two was approved by the King's College of London Research Ethics Committee. All participants provided written informed consent and were paid for participating.

## Additional files

### Supplementary files

• Source code 1. Gambling proportion for the different conditions of experiment one. Rows indicate participants and columns indicate conditions (from left to right: £2, £3, £4, £5 EV in the high-variance context, and £3 and £4 EV in the low variance context).

• Source code 2. Data from experiment two. Rows indicate participants and columns indicate data. The first twelve columns report the gambling proportion for the different conditions (from left to right: HV-deck-set, £9 deck, £11 card; HV-deck-set, £9 deck, £9 card; HV-deck-set, £9 deck, £7 card; HV-deck-set, £7 deck, £9 card; HV-deck-set, £7 deck, £7 card; HV-deck-set, £7 deck, £5 card; LV-deck-set, £7 deck, £9 card; LV-deck-set, £7 deck, £7 card; LV-deck-set, £7 deck, £5 card; LV-deck-set, £5 deck, £7 card; LV-deck-set, £5 deck, £5 card; LV-deck-set, £5 deck, £3 card). Columns 13 and 14 report the average gambling proportion and the effect of card amount on gambling probability (i.e., i.e., the associated regression coefficient of the logistic regression model), respectively. For each deck-set, the difference in gambling between lower and higher value decks for amounts common to both decks (corresponding to £5 and £7 in the low-value deck-set, and to £7 and £9 in the high-value deck-set) was computed and the mean of these two differences is reported in column 15. Column 16 reports the difference in gambling between the low and high value deck-set for the £7 deck (common to both deck-sets).

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

## Appendix

We derive *equation 7* from the generative model shown in *Figure 1C*. A higher-level contextual variable (e.g., the neighbourhood) is represented by a Gaussian distribution with mean $\mu_{HC}$ equal to zero and variance $\sigma_{HC}^2$, from which a value HC is sampled. Sensory evidence about HC is provided and represented by HO which is sampled from a Gaussian distribution with mean HC and variance $\sigma_{HO}^2$. A lower-level contextual variable (e.g., the restaurant) is represented by a (Gaussian) distribution with mean HC and variance $\sigma_{LC}^2$, from which a value LC is sampled. Sensory evidence about LC is provided and represented by LO which is sampled from a Gaussian distribution with mean LC and variance $\sigma_{LO}^2$ A reward is obtained and sampled from a Gaussian distribution with mean LC and variance $\sigma_R^2$. We propose that agents infer the posterior distribution P(LC|HO,LO,R) sequentially by estimating, in the order, P(HC|HO), P(LC|HO), P(LC|HO,LO), and P(LC|HO,LO,R). The posterior mean of P(HC|HO) is:

$$\mu_{HC|HO} = \frac{\sigma_{Hc}^2}{\sigma_{HC}^2 + \sigma_{HO}^2} HO \tag{12}$$

And the posterior variance:

$$\sigma_{HC|HO}^2 = \sigma_{HC}^2 - \frac{\sigma_{HC}^2}{\sigma_{HC}^2 + \sigma_{HO}^2} \sigma_{HC}^2 \tag{13}$$

The posterior mean of P(LC|HO) is equal to $\mu_{HC|HO}\left(\mu_{LC|HO} = \mu_{HC|HO}\right)$ , while the posterior variance is:

$$\sigma_{LC|HO}^2 = \sigma_{HC|HO}^2 + \sigma_{LC}^2 \tag{14}$$

The posterior mean of P(LC|HO,PO) is:

$$\mu_{LC|HO,LO} = \mu_{LC|HO} + \frac{\sigma_{LC|HO}^2}{\sigma_{LC|HO}^2 + \sigma_{LO}^2}\left(LO - \mu_{LC|HO}\right) \tag{15}$$

And the posterior variance:

$$\sigma_{LC|HO,LO}^2 = \sigma_{LC|HO}^2 - \frac{\sigma_{LC|HO}^2}{\sigma_{LC|HO}^2 + \sigma_{LO}^2} \sigma_{LC|HO}^2 \tag{16}$$

Finally, the posterior mean of P(LC|HO,LO,R) is:

$$\mu_{LC|HO,LO,R} = \mu_{LC|HO,LO} + \frac{\sigma_{LC|HO,LO}^2}{\sigma_{LC|HO,LO}^2 + \sigma_R^2}\left(R - \mu_{LC|HO,LO}\right) \tag{17}$$

Implying (with few algebraic transformations) the following incentive value for the reward:

$$V(R) = \frac{\sigma_{LC|HO,LO}^2}{\sigma_{LC|HO,LO}^2 + \sigma_R^2}\left(R - \frac{\sigma_{LC|HO}^2}{\sigma_{LC|HO}^2 + \sigma_{LO}^2}LO - \frac{\sigma_{LO}^2}{\sigma_{LC|HO}^2 + \sigma_{LO}^2}\frac{\sigma_{HC}^2}{\sigma_{HC}^2 + \sigma_{HO}^2}HO\right) \tag{18}$$

This equation implements three normalization factors: (i) a subtractive normalization factor $\left( \frac{\sigma^2_{LC|HO}}{\sigma^2_{LC|HO}+\sigma^2_{LO}} LO \right)$ proportional to the value LO observed at the low contextual level, (ii) a subtractive normalization factor $\left( \frac{\sigma^2_{LO}}{\sigma^2_{LC|HO}+\sigma^2_{LO}} \frac{\sigma^2_{HC}}{\sigma^2_{HC}+\sigma^2_{HO}} HO \right)$ proportional to the value HO observed at the high contextual level, (iii) a divisive normalization factor $\left( \frac{\sigma^2_{LC|HO,LO}}{\sigma^2_{LC|HO,LO}+\sigma^2_{R}} \right)$ that captures the weighting dependent on the (relative) reward variance. If we define the three factors as and and K respectively, we obtain **equation 7**.

