## [Decision Letter]

Thank you for submitting your article "A Bayesian model of context-sensitive value attribution" for consideration by *eLife*. Your article has been favorably evaluated by Sabine Kastner (Senior Editor) and three reviewers, one of whom, Sam Gershman (Reviewer #1), also served as Guest Reviewing Editor.

The reviewers have discussed the reviews with one another and the Reviewing Editor has drafted this decision to help you prepare a revised submission.

Summary:

This work develops a Bayesian theory of context effects in decision making, and tests it in a behavioral experiment. The proposed model goes beyond previous efforts by introducing multiple levels of hierarchy. The reviewers agree that this is an interesting and important topic, but have concerns about the analysis methodology, presentation of the model/results, adequacy of the experimental data, and relation to previous work.

Essential revisions:

1) Analysis methodology: please address the points raised by Reviewers 1 and 3 concerning statistical analyses and model comparison.

2) Presentation of the model/results: all the reviewers suggested ways in which the presentation could be improved, including clarification of modeling details and task description.

3) Adequacy of experimental data: Reviewer 2 raised important concerns about whether the reported experiment provides a strong test of the theory (particularly the precision-weighting component), and Reviewer 3 pointed out a number of confounds in the design. The reviewers are in agreement that the paper should include new experimental data that addresses the confounds and ideally also provides a stronger test of the theory.

4) Relation to previous work: Reviewer 1 points out several other theoretical frameworks for certain context effects. Please address these in the revision.

Reviewer #1:

1) I didn't see what would seem to be the most obvious analysis of the choice data, namely comparing the LV vs. HV bars (as well as 5 vs. 9 bars) in Figure 4. The authors report correlations but shouldn't the model predict a difference in means as in Figure 2?

2) The simulation details were unclear to me. Did the authors simulate a synthetic dataset many times and then average the correlations, or did they just run a single simulation? I think the former procedure is better justified, since we don't know whether the results found with the simulations are idiosyncratic or reliable.

3) Some researchers, such as Rangel & Clithero (2012) and Louie, Glimcher & Webb (2015), have drawn connections between context effects and efficient coding, appealing to the idea that divisive normalization is a mechanism for removing statistical redundancies. A Bayesian theory like BCV enables but does not require efficient coding; a given distribution could be coded with varying degrees of efficiency. However, BCV also appears to make some claims about mechanistic implementation which might be relevant to the question of efficiency. How do the authors see the relationship between these theories and BCV? More generally, the authors could deepen their contribution by considering the realization of BCV across Marr's levels.

4) Showing that BCV can account for earlier value normalization results would also bolster the theory. In addition, there is a rich literature on context/decoy effects that would be relevant to at least mention here. Relatedly, work in economics has studied the idea that reference points depend on expectations (e.g., Koszegi & Rabin, 2006), and there have been a number of important recent papers on range effects and relativistic choice processes (e.g., Bordalo et al., 2012; Cunningham, 2013,; Bushong, Rabin & Schwartzstein, 2015). It would be illuminating to better understand how these frameworks relate to BCV.

Reviewer #2:

My major reservation about the paper is that the human data offers only partial confirmation for the Bayesian context model. In its full specification (in the Appendix), the BCV model predicts that incentive value will depend on a divisive scaling term (implementing precision weighting dependent on relative reward variance) and subtractive prediction error terms (low and high context reward predictions, weighted by their relative contextual cue variances). However, the effects of precision – which are a key element of Bayesian approaches – are untested in the analysis of the experimental data. As the authors state in the paper, reward variance is not manipulated and the divisive term (K) cannot be examined. In addition, the subtractive terms should also be precision-weighted by terms relating posterior and prior variances (denoted by tau_LO and tau_HO). However, the results only speak to overall context effects, essentially asking whether there is an overall effect of low and high context on gambling (population regression effects in Figure 5, and the chi-square tests showing positive tau_LO and tau_HO parameters).

The issue is that adaptive effects to average rewards are well known (for example, successive contrast effects in the animal literature and reference point models like prospect theory); without validating the precision-dependent predictions of their model, I'm not sure that the authors can convincingly argue that BCV is a more appropriate model – particularly as other models are not tested. Given the experimental setup, testing the divisive weighting term is not possible in this dataset; however, can the authors make any predictions about not just the significance but the relative magnitude of the weighting factors tau_LO and tau_HO (predicted and fit to data)?

Reviewer #3:

1) Behavioural effects of reward context on incentive value are already well-established and even the specific paradigm used here has already been published by the same authors. Is there any new, surprising behavioural effect that follows from the new model?

2) The Bayesian model presented in the manuscript is not formally compared to other well-established models that may similarly account for the behavioural effects. The authors should show with formal model comparisons that their model outperforms other classic (non-Bayesian) models commonly employed to model context effects on value-based choice.

3) The model is not biologically realistic. This is not always a problem; in fact, there are many elegant demonstrations that Bayesian frameworks can account for optimal performance in various domains better than other accounts. However, in the specific context of reward-guided decision-making, it is unclear why and how a Bayesian framework should apply, and to what degree it is more consistent with behaviour and the underlying neural computations. Please provide a lot more information on how this model may be implemented by neural computations. In particular, it would help if there was any empirical evidence for the hierarchical representation of reward context.

4) The Abstract claims that the model "generates new empirical predictions and may help explain important phenomena in psychopathologies such as addiction." I found the corresponding text in the Discussion rather vague. Please provide explicit predictions for specific experimental effects that follow from this model and please explain much more concretely which important phenomena in psychopathologies are explained by it.

5) The manuscript claims in several places that a reward's incentive value corresponds to the (precision-weighted) prediction error. This is misleading. By definition, the incentive value is the property of a stimulus/expected reward that triggers approach behaviour and choice of the corresponding option. This representation must therefore be computed before the choice is taken and the reward is obtained. The prediction error, by contrast, is the deviation of the reward obtained as a consequence of the choice from the reward expected prior to the choice. This post-choice representation can therefore not be the incentive value guiding choice. The authors need to clarify their terminology and ensure that they remain consistent with established definitions in the literature.

In addition to the conceptual points listed above, the manuscript also has shortcomings with respect to methodology and results presentation that will need to be addressed:

6) The behavioural task was not designed to allow proper tests of the full model. Some of these problems are listed by the authors themselves and have led to adaptations of the model so that it could be fit to the data. For instance, reward variance in the different decks is not varied, is heavily constrained (there are only 3 different reward values per deck), and is perfectly correlated with average reward magnitude. To properly test whether the precision of reward prediction errors established by the different contexts really plays an important role, the authors should fit the full model to datasets with contexts that differ substantially in their reward variance and that disentangle reward magnitude from reward variance. Moreover, the contexts should be associated with a lot more than just three possible reward values so that the form of the expected reward distributions can be properly approximated (see below).

7) The model specification does not match the environment established by the behavioural task. The decks were associated with 3 equiprobable values per deck. Therefore, "smart" subjects would employ a flat discrete expectation of the three possible values within the given context. In contrast, the modelling solution is based on continuous (Gaussian) distributions that are not restricted by the bounds imposed by each context. There are two problems with this: (1) If we assume that the subjects indeed optimally integrate all information, then a flat prior belief bounded by the context's minimum and maximum reward would be accurate. This cannot be modelled by the presented specification. (2) Even if subjects employed continuous Gaussian priors to model reward expectations, such distributions would probably not be narrow enough with respect to the context bounds (i.e., the priors would wrongly lead to expectations of rewards that are outside of the bounded scale). These problems will probably be evident if the authors report the values of the latent variables after fitting the model to the empirical data. Please include such a table to allow the reader to inspect this issue.

In my view, if the authors really wanted to maintain a Bayesian optimal observer model, then they should examine how the prior (potentially flat, but can also be modelled if the authors like) is combined with the likelihood of the actual numeric representation to obtain a posterior estimate that should naturally occur within the actual numeric bounds of the context (for an example on how to formally deal with Bayesian problems of this kind, i.e. bounded contexts, see Jazayeri and Shadlen 2010, Nature Neuroscience). This formal specification can then be expanded to the interesting contextual hierarchical framework that the authors propose in their study.

8) The authors rescaled the contextual averages and reward values to perform their model fits. Why? A correctly specified model should be able to take as inputs the actual values of the contexts and rewards of their behavioural paradigm (which are all single-digit numbers after all). This would help to assess the model's explanatory power.

9) The authors perform model comparison by summing log-likelihoods across participants. I find the selection of this approach for model comparison surprising, given that several of the co-authors have pushed the use of precise Bayesian model selection methods that properly account for the complexity and variability of the model fits across trials and participants. The authors should employ such methods and should provide values quantifying the quality of the model fits after penalizing for model complexity.

10) The behavioural task is described in a fashion that makes it hard to replicate. For instance, is it true that on every trial, a card was drawn from the blue deck? If so, how were the different deck contexts varied across the different colours? Please make sure the task described in sufficient detail so that another person could program it.

---

## [Author Response]

*Essential revisions:*

1) Analysis methodology: please address the points raised by Reviewers 1 and 3 concerning statistical analyses and model comparison.

We believe that all the points concerning statistical analyses and model comparison have been addressed satisfactorily. Please see specific replies to reviewers below.

2) Presentation of the model/results: all the reviewers suggested ways in which the presentation could be improved, including clarification of modeling details and task description.

We are grateful for the feedback from reviewers on this aspect, and we have clarified the presentation of the model and task description (see replies below).

3) Adequacy of experimental data: Reviewer 2 raised important concerns about whether the reported experiment provides a strong test of the theory (particularly the precision-weighting component), and Reviewer 3 pointed out a number of confounds in the design. The reviewers are in agreement that the paper should include new experimental data that addresses the confounds and ideally also provides a stronger test of the theory.

Thanks for the feedback on this point. We have clarified why we believe that the experiment is adequately designed to investigate a key prediction of BCV in relation to contextual influences at multiple hierarchical levels. However, we also agree that the previous experiment is not suitable to test other key predictions of the theory; most importantly the effect of precision as suggested by reviewers.

Investigating the role of precision is something we already had in mind when we had completed this study. Indeed, we already had run an experiment investigating the role of context precision in value adaptation. As the reviewers have highlighted this point we are delighted to be able to include these new data in our revision. These data provide supportive evidence for a role of context precision, consistent with BCV. As pointed out by the reviewers, such role is a key feature of Bayesian schemes and is something which specifically supports BCV.

In sum, we now report a new experiment (along the lines suggested by the reviewers) investigating the role of reward variance (experiment one) together with the previous experiment investigating the role of hierarchical contexts (experiment two).

4) Relation to previous work: Reviewer 1 points out several other theoretical frameworks for certain context effects. Please address these in the revision.

Thanks for feedback on this point. We now link our work more extensively with other theoretical frameworks (and we compare explicitly predictions from different frameworks; see replies below).

*Reviewer #1:*

1) I didn't see what would seem to be the most obvious analysis of the choice data, namely comparing the LV vs. HV bars (as well as 5 vs. 9 bars) in Figure 4. The authors report correlations but shouldn't the model predict a difference in means as in Figure 2?

Thanks for the feedback on this point, and we agree this needs to be explained better. Figure 2 shows predictions about incentive value V(R) postulated by BCV in the different conditions of our experiments. Crucially, the predicted V(R) does not correspond to the predicted gambling proportion, as (based on previous findings; Rigoli et al., 2016a; 2016b) we expected the gambling proportion to depend on V(R) in a different way for different participants. Specifically, we predicted that the incentive value determines choice behaviour according to a logistic regression P(gambling) = sigmoid(α V(R)+ρ), where the slope α is positive or negative for different participants. Given this prediction and the adaptation processes postulated by BCV: if we consider two contexts with common trial EVs, we expected that participants with positive α would gamble more for common EVs when these are associated with larger V(R), while we expected that participants with negative α would gamble more for common EVs when these are associated with smaller V(R). To test these predictions we correlated α (i.e., the slope of the logistic regression) with the difference in gambling for common EVs. Note that these predictions do not imply an overall difference in mean gambling across contexts, as the gambling difference across contexts is expected to differ when comparing participants with positive and negative α. This issue is now clarified in analysis of experiment one:

“Though we observed no overall difference in gambling across participants for £4 and £3 EV (see above), participants could be differentiated based on those who gambled more with £4 or £3 EV. […] To examine these predictions we tested for an interaction, corresponding to the differential gambling percentage across contexts (low-variance minus high-variance context) for £4 choices minus the differential gambling percentage across contexts (low-variance minus high-variance context) for £3 choices.”

In analysis of experiment two:

“Though across participants we observed no overall effect of option EV on gambling, participants could be differentiated based on those who showed a positive or negative effect of option EV on gambling. […] We investigated this hypothesis both at the level of decks and deck-sets (Figure 6).”

And in the caption of Figure 6:

“Gambling proportion for different card amounts and different context conditions, separately for (A) participants showing a negative effect of card monetary amount (i.e., the slope of the logistic regression model with card amount as predictor) on gambling (n = 19) and (B) participants showing a positive effect of card amount on gambling (n = 13). […] These data are consistent with our hypotheses; namely (i) with predictions arising from BCV (Figure 2) which implies subtractive normalization of incentive value at both hierarchical levels and (ii) with the prediction (derived from previous observations; Rigoli et al., 2016a; 2016b) that the influence of incentive value on gambling proportion depends on the individual preference to gamble with large or small card amounts.”

2) The simulation details were unclear to me. Did the authors simulate a synthetic dataset many times and then average the correlations, or did they just run a single simulation? I think the former procedure is better justified, since we don't know whether the results found with the simulations are idiosyncratic or reliable.

The referee is correct and in the initial submission of the manuscript a single simulation was employed, but we agree that a better method would entail repeating several simulations and considering the average correlation from these simulations. We have followed this suggestion and this method is now adopted in the new manuscript, where 100 simulations where considered and the average correlation reported. This is now clarified:

“Finally, we used the (selected) model and subject-specific parameter estimates of the last analysis to generate simulated choice behaviour and performed behavioural analyses on the ensuing data (data were simulated 100 times and the average statistics are reported).”

3) Some researchers, such as Rangel & Clithero (2012) and Louie, Glimcher & Webb (2015), have drawn connections between context effects and efficient coding, appealing to the idea that divisive normalization is a mechanism for removing statistical redundancies. A Bayesian theory like BCV enables but does not require efficient coding; a given distribution could be coded with varying degrees of efficiency. However, BCV also appears to make some claims about mechanistic implementation which might be relevant to the question of efficiency. How do the authors see the relationship between these theories and BCV? More generally, the authors could deepen their contribution by considering the realization of BCV across Marr's levels.

Thanks for this suggestion for greater elaboration on this point. We agree a crucial aspect of any theory of value adaptation needs to address implications for efficient coding. This is now discussed here, together with a more general framing of the theory within the Marr’s model, as proposed by the reviewer:

“With reference to the three levels of analysis (i.e., computational, algorithmic and implementation) proposed by Marr (1982), BCV speaks to the computational level as it focuses on normative principles (implicit in optimal Bayesian inference) proposed to explain value and choice adaptation. […] In other words, BCV implies that, as well as neural signalling, behaviour itself is tuned to the statistics of the incentives, so as to maximize discriminability among these incentives.”

*4) Showing that BCV can account for earlier value normalization results would also bolster the theory. In addition, there is a rich literature on context/decoy effects that would be relevant to at least mention here. Relatedly, work in economics has studied the idea that reference points depend on expectations (e.g., Koszegi & Rabin, 2006), and there have been a number of important recent papers on range effects and relativistic choice processes (e.g., Bordalo et al., 2012*; *Cunningham, 2013; Bushong, Rabin & Schwartzstein, 2015). It would be illuminating to better understand how these frameworks relate to BCV.*

We agree it is relevant to discuss the theory as it relates to previous models mentioned by the reviewer (and other classical theories). We now do this:

“The proposal advanced here has some similarities with classical theories of value, such as Expected Utility theory (von Neumann & Morgenstern, 1944) and Prospect theory (Kahneman & Tversky, 1979). […] For instance, BCV predicts that divisive normalization derives from the gain term (i.e., reward variance) and not from the expected reward (Louie et al., 2014; 2015), and that the reward variance divides – and not multiplies (Kőszegi & Szeidl, 2014) – the prediction error. Importantly, these predictions are not ad hoc but derive necessarily from Bayesian assumptions.”

In addition, we agree it is important to discuss the theory in light of earlier empirical results. As also indicated by the reviewer, most empirical studies have focused on effects elicited by options simultaneously available during choice and where multiple attributes are presented (e.g. the decoy effect). On the contrary, BCV focuses on context effects elicited by expectations before options are presented and where a single attribute is salient. Data on this domain are relatively scarce, and are discussed in the introduction to the new experiment we now report (experiment one in the new manuscript):

“Data from conditions where BCV is applicable, namely those involving a single attribute and where context depends on past options (and not simultaneously presented options), are relatively scarce. […] Here, we present data from a behavioural experiment where we investigate this very question.”

We also mention explicitly other forms of context effect emerging in conditions where BCV is not applicable at the moment, like the decoy effect. We acknowledge this as a shortcoming of BCV, though we argue that similar principles might be valid in these conditions too:

“We highlight shortcomings of the model, though the framework itself may be fruitful in addressing some of these shortcomings. […] This would provide an opportunity to model attentional processes determining an optimal weighting of different attributes based on their importance and reliability.”

*Reviewer #2:*

*My major reservation about the paper is that the human data offers only partial confirmation for the Bayesian context model. In its full specification (in the Appendix), the BCV model predicts that incentive value will depend on a divisive scaling term (implementing precision weighting dependent on relative reward variance) and subtractive prediction error terms (low and high context reward predictions, weighted by their relative contextual cue variances). However, the effects of precision – which are a key element of Bayesian approaches – are untested in the analysis of the experimental data. As the authors state in the paper, reward variance is not manipulated and the divisive term (K) cannot be examined. In addition, the subtractive terms should also be precision-weighted by terms relating posterior and prior variances (denoted by tau_LO and tau_HO). However, the results only speak to overall context effects, essentially asking whether there is an overall effect of low and high context on gambling (population regression effects in Figure 5, and the chi-square tests showing positive tau_LO and tau_HO parameters).*

The issue is that adaptive effects to average rewards are well known (for example, successive contrast effects in the animal literature and reference point models like prospect theory); without validating the precision-dependent predictions of their model, I'm not sure that the authors can convincingly argue that BCV is a more appropriate model – particularly as other models are not tested. Given the experimental setup, testing the divisive weighting term is not possible in this dataset; however, can the authors make any predictions about not just the significance but the relative magnitude of the weighting factors tau_LO and tau_HO (predicted and fit to data)?

We are grateful for this suggestion, and agree on the importance of demonstrating an actual effect of precision. Such a test is something we already had in mind when preparing this manuscript, and hence, before obtaining these reviews, we had run (and analysed data from) a novel experiment investigating the role of context precision in value adaptation. We are grateful that the reviewer has highlighted the importance of this point – and now we present data from this new experiment which addresses this very point (experiment one in the new manuscript). In brief, this provides evidence for context precision consistent with BCV. As pointed out by the reviewer, such a role is a key feature of Bayesian schemes and is an observation which supports BCV over alternative theories (e.g., those based on reference point adaptation alone such as Prospect theory).

The question about the comparison between factors tau_LO and tau_HO is interesting, though we believe that our task is not suitable to address this question. An experiment aimed at investigating this question will require an explicit manipulation of the salience of information regarding different hierarchical levels (e.g., manipulating the relative “perceptual noise” of this information), while in our task it is unclear whether (and which) level is more salient.

Reviewer #3:

1) Behavioural effects of reward context on incentive value are already well-established and even the specific paradigm used here has already been published by the same authors. Is there any new, surprising behavioural effect that follows from the new model?

We agree this should be made clearer. Two novel and important empirical predictions generated by BCV are: (i) the fact that hierarchically organized contexts can exert adaptation on incentive value and choice, (ii) the variance of the reward context can exert adaptation on incentive value and choice. We investigate both predictions in the current version of the manuscript. The first prediction is tested in the behavioural study reported also in initial submission version of the manuscript. In the present version, we now report a new experiment where we also test the second prediction. These issues are now clarified in the Discussion:

“We propose a Bayesian scheme (BCV) as a model of contextual influences on incentive value attribution. BCV is based on Bayesian inference principles and on generative models of reward. Adopting two novel experimental designs, we provide behavioural evidence that supports two key predictions of BCV, namely that value attribution is affected by reward variance (which exerts divisive normalization) and by hierarchically organized contexts.”

2) The Bayesian model presented in the manuscript is not formally compared to other well-established models that may similarly account for the behavioural effects. The authors should show with formal model comparisons that their model outperforms other classic (non-Bayesian) models commonly employed to model context effects on value-based choice.

Thanks for highlighting this point. We agree that the comparison with other theories should be explained better and formally tested. We now do this in the new version of the manuscript. The connections with other theories are explained:

“The proposal advanced here has some similarities with classical theories of value, such as Expected Utility theory (von Neumann & Morgenstern, 1944) and Prospect theory (Kahneman & Tversky, 1979). […] Importantly, these predictions are not ad hoc but derive necessarily from Bayesian assumptions.”

In addition, we perform formal model comparison to compare BCV with predictions derived from alternative theories. This is done for experiment one:

“We next compared the full model with an alternative model without a subtractive normalization component (as postulated by BCV, where the expected reward is subtracted to the actual reward); namely, where the incentive value was equal to V(R)=τεR. […] This test was significant (χ^2^(1) = 508, p < 0.001), meaning that the model implementing both subtractive and divisive normalization (derived from BCV and described by [Disp-formula equ8]) fits the data better.”

And for experiment two:

“We next compared the full model with an alternative model where the context parameters (capturing the influence of the reward expected within a context at multiple hierarchical levels) divided the reward rather than being subtracted from the reward; in other words where the incentive value corresponds to:(11)V(R)=R1+χLOτLO+χHOτHO

[…] This test was significant (full models: χ^2^(1) = 78, p < 0.001; for models with α, ρ and τLO: (1) = 26, p < 0.001; for models with α, ρ and τ_*LH*_: (1) = 50, p < 0.001), meaning that the model implementing subtractive normalization (consistent with BCV) is a better explanation for the data.”

3) The model is not biologically realistic. This is not always a problem; in fact, there are many elegant demonstrations that Bayesian frameworks can account for optimal performance in various domains better than other accounts. However, in the specific context of reward-guided decision-making, it is unclear why and how a Bayesian framework should apply, and to what degree it is more consistent with behaviour and the underlying neural computations. Please provide a lot more information on how this model may be implemented by neural computations. In particular, it would help if there was any empirical evidence for the hierarchical representation of reward context.

Thanks for highlighting this point. We agree it is important to clarify the link between BCV and brain function. We stress that there are several existing proposals as to how Bayesian inference schemes (and hence BCV) might be implemented in the brain (e.g., Doya et al., 2007; Friston, 2005; Hennequin et al., 2014; Knill & Pouget, 2004). Therefore we consider that BCV is relevant for ideas regarding neural implementation level, and is equally biologically plausible as other Bayesian inference schemes.

Several neural observations are consistent with BCV, such as the observation that a reward signal in several brain regions is adapted to a reference point and to reward range (Bermudez & Schultz, 2010; Cox & Kable, 2014; Louie et al., 2011; Padoa-Schioppa, 2009; Padoa-Schioppa & Assad, 2008; Park et al., 2012; Rigoli et al., 2016a; Tobler et al., 2005 Kobayashi et al., 2010; Tremblay & Schultz, 1999), and like a recent finding of a link between neural and choice adaptation (Rigoli et al., 2016). Yet there are important aspects of BCV which will need to be probed at the neural level. For instance, as pointed out by the reviewer, a test of an hypothesis of a hierarchical organization of reward context in the brain is lacking. However, we believe that the fact that aspects of the model have not yet been tested represents a strength (rather than a weakness) of the theory, as it makes novel and precise predictions. We now refer to these issues in the following text:

“With reference to the three levels of analysis (i.e., computational, algorithmic and implementation) proposed by Marr (1982), BCV speaks to the computational level as it focuses on normative principles (implicit in optimal Bayesian inference) proposed to explain value and choice adaptation. […] In other words, BCV implies that, as well as neural signalling, behaviour itself is tuned to the statistics of the incentives, so as to maximize discriminability among these incentives.”

4) The Abstract claims that the model "generates new empirical predictions and may help explain important phenomena in psychopathologies such as addiction." I found the corresponding text in the Discussion rather vague. Please provide explicit predictions for specific experimental effects that follow from this model and please explain much more concretely which important phenomena in psychopathologies are explained by it.

We agree this was not clear. Two empirical predictions arising from the theory are: (i) hierarchically organized contexts exert adaptation of incentive value and choice, (ii) variance of the reward context can exert adaptation of incentive value and choice. We investigate both predictions here, the second with new experimental data. The first prediction was tested in the previous version of the manuscript. In this revised manuscript, we report an experiment where the second prediction is tested. The specific predictions of the model are now presented in detail in Figure 2 and are discussed here in relation with other models:

“The proposal advanced here has some similarities with classical theories of value, such as Expected Utility theory (von Neumann & Morgenstern, 1944) and Prospect theory (Kahneman & Tversky, 1979). […] Importantly, these predictions are not ad hoc but derive necessarily from Bayesian assumptions.”

In relation with the link with psychopathology, we focus on addiction. BCV may be useful to understand the computational mechanisms underlying the development of addiction and tolerance. In addition, BCV may be useful also to clarify the computational mechanisms representing an individual predisposition to drug abuse, for instance dependent on increased uncertainty of prior belief on reward. This is clarified:

“Finally, there are questions related to psychopathology that can be fruitfully formulated in terms of BCV, for example addiction. […] For instance, increased uncertainty over prior reward beliefs may boost the magnitude of the (positive) prediction error elicited by drug consumption, hence enhancing individual predisposition to drug addiction.”

5) The manuscript claims in several places that a reward's incentive value corresponds to the (precision-weighted) prediction error. This is misleading. By definition, the incentive value is the property of a stimulus/expected reward that triggers approach behaviour and choice of the corresponding option. This representation must therefore be computed before the choice is taken and the reward is obtained. The prediction error, by contrast, is the deviation of the reward obtained as a consequence of the choice from the reward expected prior to the choice. This post-choice representation can therefore not be the incentive value guiding choice. The authors need to clarify their terminology and ensure that they remain consistent with established definitions in the literature.

We apologise for the confusion and agree on a need for clarity. BCV is applicable every time novel information about reward is provided, which is when a prediction error occurs. This happens: (i) when a (primary or secondary) reward is delivered (or is not delivered when expected), which can be post choice but also in other conditions (e.g., in classical conditioning paradigms), when a reward is delivered independent of action) (ii) when one (or more) option is presented, which is the condition exploited in our experiments. This is because an agent has expectancies about the value of an option, which can be confirmed or not when options are presented, leading to prediction errors. We agree that, to avoid confusion, this needs to be clarified. We stress that the idea that a prediction error occurs when options are presented is not uncommon in the reward literature, as in standard computational temporal difference algorithms (Sutton & Barto, 1998). This is now clarified:

“Equation two applies every time novel information about reward is provided, which is when a prediction error occurs. […] The latter follows because an agent has an expectation about an option, which leads to a prediction error when the actual option is presented.”

*In addition to the conceptual points listed above, the manuscript also has shortcomings with respect to methodology and results presentation that will need to be addressed:*

6) The behavioural task was not designed to allow proper tests of the full model. Some of these problems are listed by the authors themselves and have led to adaptations of the model so that it could be fit to the data. For instance, reward variance in the different decks is not varied, is heavily constrained (there are only 3 different reward values per deck), and is perfectly correlated with average reward magnitude. To properly test whether the precision of reward prediction errors established by the different contexts really plays an important role, the authors should fit the full model to datasets with contexts that differ substantially in their reward variance and that disentangle reward magnitude from reward variance. Moreover, the contexts should be associated with a lot more than just three possible reward values so that the form of the expected reward distributions can be properly approximated (see below).

We appreciate this suggestion, and agree that investigating the effect of precision is a fundamental test for BCV. Such test is something we already had in mind, and had run a novel experiment prior to submission of the original manuscript. This experiment is relevant as in it we investigate the role of context precision in value adaptation. We now include these data (experiment one in the new manuscript) which provide evidence for a role of context precision consistent with BCV. As pointed out by the reviewer, such role is a key feature of Bayesian schemes and supports BCV over alternative accounts.

To avoid confusion, we now clarify in the novel paradigm that there are two types of variance. The first refers to the variance of possible outcomes of the gamble (which is perfectly correlated with the EV of options as in a previous experiment and as in (Rigoli et al., 2016a; 2016b), and is not the focus of the experiment. The second refers to the variance across options (i.e., the variance characterizing the distribution of serially-provided options), which is what we manipulate and is reflected in and affects the gain term of equation one and two. This is clarified:

“Note that in our task there are two types of variance. The first refers to the variance of possible outcomes of the gamble (which is perfectly correlated with the EV of options, as in Rigoli et al., 2016), and is not the focus of our study. The second refers to variance across options (i.e., the variance characterizing the distribution of successive options), which is what we experimentally manipulate and investigate. In the model, this is reflected in and affects the gain term in [Disp-formula equ2].”

7) The model specification does not match the environment established by the behavioural task. The decks were associated with 3 equiprobable values per deck. Therefore, "smart" subjects would employ a flat discrete expectation of the three possible values within the given context. In contrast, the modelling solution is based on continuous (Gaussian) distributions that are not restricted by the bounds imposed by each context. There are two problems with this: (1) If we assume that the subjects indeed optimally integrate all information, then a flat prior belief bounded by the context's minimum and maximum reward would be accurate. This cannot be modelled by the presented specification. (2) Even if subjects employed continuous Gaussian priors to model reward expectations, such distributions would probably not be narrow enough with respect to the context bounds (i.e., the priors would wrongly lead to expectations of rewards that are outside of the bounded scale). These problems will probably be evident if the authors report the values of the latent variables after fitting the model to the empirical data. Please include such a table to allow the reader to inspect this issue.

In my view, if the authors really wanted to maintain a Bayesian optimal observer model, then they should examine how the prior (potentially flat, but can also be modelled if the authors like) is combined with the likelihood of the actual numeric representation to obtain a posterior estimate that should naturally occur within the actual numeric bounds of the context (for an example on how to formally deal with Bayesian problems of this kind, i.e. bounded contexts, see Jazayeri and Shadlen 2010, Nature Neuroscience). This formal specification can then be expanded to the interesting contextual hierarchical framework that the authors propose in their study.

We understand the reviewer’s concerns and agree. It is true that real agents may build non-Gaussian representations, especially in conditions where the true distributions are non-Gaussian. However, the advantage of using Gaussian assumptions is that posterior probability can be estimated analytically with equations which are simple and clear, and resemble common formulations widely used in decision-making and motivation research (e.g., Rescorla-Wagner update rule). Therefore, we would like to stick with Gaussian variables when presenting the model, which we believe is a better approach in making the model clear and transparent. Also on the question of fitting the model to choice data, we believe that Gaussian assumptions should not be problematic, as the equations used are more general. In other words, these equations can be derived from Gaussian assumptions but are valid also for other distributions such as uniform distributions (in general, non-skewed distributions).

Please consider [Disp-formula equ8]:V(R)=τε(R−μC)

This simply prescribes a subtractive normalization to the average reward (R−μC), and a divisive normalization elicited by reward variance, captured by a quantity τ (a free parameter bounded between 0.1 and 10) in the low-variance context (where ε = 1 indicates a low variance and ε = 0 indicates high-variance context). This equation is valid also for other distributions, such as uniform distributions (in general, non-skewed distributions). In addition, we did not impose a bound a priori on the parameter τ between 0 and 1, as would be predicted by BCV. This because we tested whether the τ estimated from choice data was between 0 and 1.

Similarly, please consider [Disp-formula equ10]:V(R)=R−χLOτLO− χHOτHO

This prescribes subtractive normalization exerted by the average reward expected at two hierarchical levels. Again this equation is general and valid also for other distributions, such as uniform distributions (in general, non-skewed distributions). As above, we did not impose a bound a priori on the parameter τ to be positive, as would be predicted by BCV. This is because we *tested* whether the τ estimated from choice data was positive.

Overall, our goal was to (i) derive equations using Gaussian assumptions, but in a general form potentially consistent also with other non-skewed distributions (e.g., uniform) (ii) fit these equations to choice with unconstrained context parameters, (iii) test whether the estimated context parameters where consistent with predictions.

The problem of context boundaries is also relevant. In general, in Bayesian approaches, it is often useful to model participants’ beliefs in continuous space, even in contexts which are “objectively” bounded. For instance, in models of normalization, incentive values go beyond the “objective” context boundaries almost by definition. Therefore we would expect that if one assumes bounded variables within the generative model, we would not obtain normalization. This we believe represents a fundamental argument for using continuous and not bounded variables in the context of value normalization. This is clarified in the Discussion:

“Here, we have assumed that variables of the generative model are Gaussian. This allows us to present the model in a simple and clear way, as posterior beliefs can be inferred analytically with relatively simple equations as adopted in standard decision-making schemes (Rescorla & Wagner, 1972). […] However, the key idea (tested in our experiments – and here derived from Gaussian assumptions) that reward average and variance elicit subtractive and divisive normalization, respectively, is quite general and can also be applied, for instance, to uniform (and, in general, non-skewed) distributions.”

8) The authors rescaled the contextual averages and reward values to perform their model fits. Why? A correctly specified model should be able to take as inputs the actual values of the contexts and rewards of their behavioural paradigm (which are all single-digit numbers after all). This would help to assess the model's explanatory power.

Thanks for pointing this out. We now follow the reviewer’s advice and use non rescaled contextual averages:

“Where χLO indicates the average option EV for the deck (for £9 deck: χLO=4.5 ; for £7 deck: χLO=3.5; for £5 deck: χLO=2.5), χHO indicates the average option EV for the deck-set (high-value deck-set: χHO=4; low-value deck-set: χHO=3), τ_*LO*_is a free parameter that mediates contextual effects at the deck level, and τHO is a free parameter that mediates contextual effects at the deck-set level (see methods).”

9) The authors perform model comparison by summing log-likelihoods across participants. I find the selection of this approach for model comparison surprising, given that several of the co-authors have pushed the use of precise Bayesian model selection methods that properly account for the complexity and variability of the model fits across trials and participants. The authors should employ such methods and should provide values quantifying the quality of the model fits after penalizing for model complexity.

Thanks for highlighting this point. It is true that in the previous version we estimated the parameters without strong constraints. This is potentially problematic because this method may lead to outlier estimates which are biased, and is a potential problem for our findings in relation with model-comparison and in relation with within-subjects t-tests over the context parameters. To address this possibility, we now adopt a Gaussian prior with mean zero (Daw, 2011) during estimation of the context parameters. This allows extreme parameter estimates only if they improve substantially the negative log-likelihood, hence avoiding possible biased extreme outliers. Note in addition that a prior mean of zero does not bias the t-tests since we tested the parameters against zero. For a sanity-check, in supplemental materials we now report frequencies of the parameter estimates (for experiment one: Figure 4—figure supplement 1; for experiment two: Figure 7—figure supplement 1). We can see that outlier estimates (i.e., scores larger or smaller than 3 SD compared to the mean) are absent.

For model comparison, we considered the probability of the data given the model and given the parameters estimated in this way. We used the likelihood ratio test because our model-comparison involves nested models. This approach is a standard and well-established method for model-comparison involving nested models (Casella & Berger, 2002; Daw, 2011). Penalization for model complexity is implicit in this test since it is based on the null hypothesis that data are generated with the simpler model, and hence the more complex model is accepted only if evidence for the data given the simpler model is below significance threshold. In addition, different levels of complexity are distinguished and penalized in a different way, thanks to the fact that the degrees of freedom of the likelihood ratio test (capturing how “strong” the evidence against the simper model needs to be to refuse the null hypothesis) increase with the number of extra parameters. The details regarding the method adopted for model fitting and model comparison are explained in detail:

“The free parameters of the models were estimated separately for each subject using *fminsearchbnd* function of the Optimization toolbox in Matlab. […] A chi-square test can be performed to estimate the probability that the observed *2d* is due to chance under the null hypothesis that data are generated by the nested model, allowing acceptance or rejection of the null hypothesis.”

*10) The behavioural task is described in a fashion that makes it hard to replicate. For instance, is it true that on every trial, a card was drawn from the blue deck? If so, how were the different deck contexts varied across the different colours? Please make sure the task described in sufficient detail so that another person could program it.*

Thank you – this is now clarified in the caption of Figure 5:

“The decks were coloured one in blue and the other in grey, indicating the selected and unselected deck respectively. Among these two decks shown on the screen, the selected deck (coloured in blue) alternated pseudo-randomly over blocks (each including 5 trials). In addition, two sets of decks alternated over longer blocks (20 trials) in a pseudo-random way”.

And in the main text:

“Among the two decks shown on the screen, the selected deck (coloured in blue) alternated pseudo-randomly over blocks (each including 5 trials). In addition, at some points during the task, decks were replaced by new decks. Two sets of decks alternated over blocks of 20 trials in a pseudo-random way.”